# The Learnability of an Unknown System From Input-Output Data

## Abstract

Artificial intelligence is transforming how scientists build models, test ideas, and make predictions. Beneath these advances lies a fundamental question: from the data we collect, when is learning possible in principle, and when is it not? We cast inference as a game between a learner, who holds a pool of candidate models, and an adversary, who holds the unknown ground-truth system. The learner observes the system and selects a candidate model to achieve one of three goals: identify the system, predict its output, or verify an input. We analyze 81 cases that arise by varying these goals, the available observations from both ground truth and candidates, and whether systems are single- or multi-valued. For each case, we prove whether universal solvability is possible. We make the connection to language identification and generation explicit and provide a case-by-case account of which results follow from earlier ideas and which use the input-output-fiber structure. By clarifying which observations make success achievable in principle, our results provide an information-theoretic guide within an idealized setting with countable spaces, exact realizability, and noiseless assumption.

## 1 Introduction

Many data-driven inference problems in science are about identifying, predicting, or verifying unknown systems given input-output data. For example, partial differential equation (PDE) discovery is about identifying an underlying governing differential equation that maps forcing functions and initial conditions to a system's response (Brunton et al., 2016; Rudy et al., 2017; Schaeffer, 2017); in early warning earthquake systems, we want to predict the map from ground tremors to the next location and magnitude of an earthquake (Al Banna et al., 2020; Allen et al., 2009; Mousavi et al., 2020); in automated theorem proving, we desire a verifiable map from a mathematical statement to either a valid proof or an indication that no proof exists in the fixed formal system (Harrison, 2009; Kaliszyk & Urban, 2014). In this paper, we call any map related to an unknown system a ground-truth map and denote it by $\mathcal{M}_*$. These ground-truth maps can be linear, nonlinear, single-valued, set-valued, or may even return the empty set for some inputs.

For any ground-truth map, we usually have one of the following goals in data-driven inference: (1) Map identification, in which the underlying governing map is sought that determines how inputs go to outputs (Rudy et al., 2017), (2) Map prediction, where one wants to take new (valid) inputs and predict the map's output (Kovachki et al., 2023), or (3) Map verification, where one wants not only to predict the map's output for a given input but also to verify whether that input is valid for the map (Hendrycks et al., 2020). Map identification is usually thought of as figuring out exactly what the map is, e.g., using data to write down a governing equation. Whereas, map prediction is about predicting the map's output, and map verification is for applications with a demand for certificates of success. For example, PDE discovery has the goal of map identification as one wants to write down the underlying PDE, earthquake systems have a map prediction goal as one hopes to take ground tremors to earthquakes (without necessarily understanding the governing equations), and proof generators are more about map verification. A practical use of map verification is input validation, where we wish to determine whether an input belongs to the admissible domain of the system. For proof generators, we want the system to be able to identify a false statement.

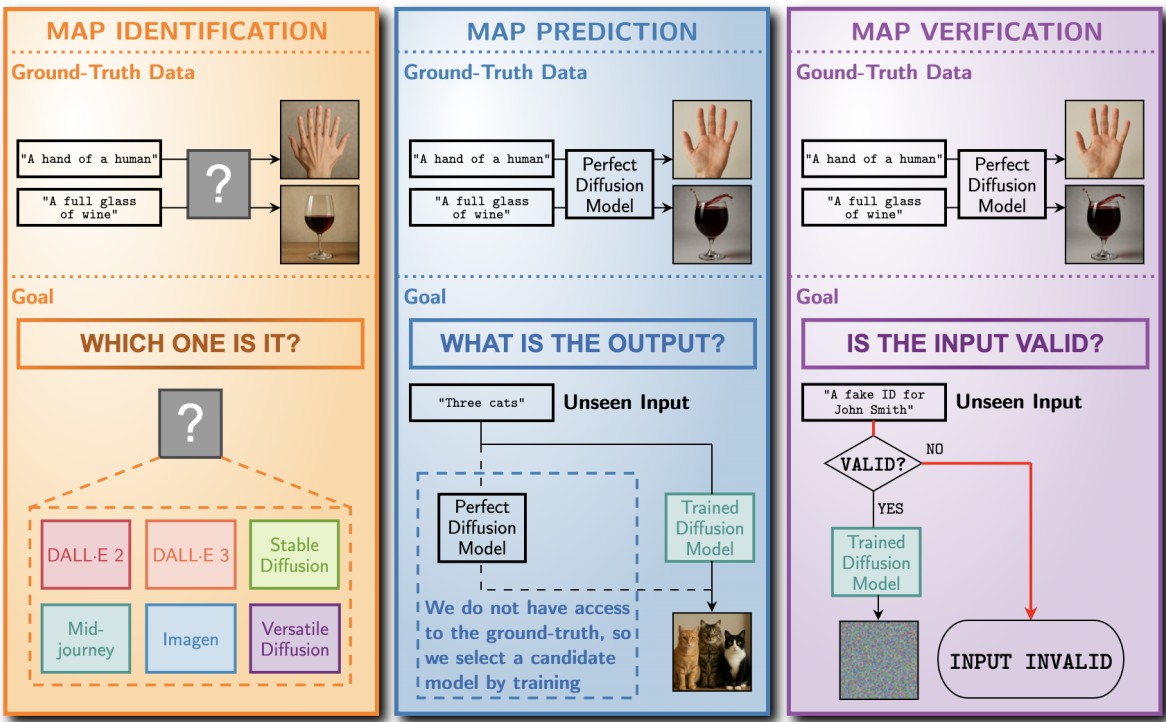

Figure 1: **Three core goals in data-driven inference, illustrated with text-to-image generation.** Each panel shows a different interaction with an unknown ground-truth map taking texts to images. Map identification (left) aims to recover the explicit mapping rules, such as identifying the ground-truth generative model producing the images from prompts. Map prediction (middle) only aims to mimic the behavior of the ground-truth map for new, previously unseen prompts. Map verification (right) is about checking whether a given image can be generated from a given text prompt, useful for input validation.[1]

The ground-truth map that one wishes to either identify, predict, or verify can come from many sources. In PDE discovery, we often start with a dictionary of possible terms for the equations, giving us a set of candidate differential equations (Brunton et al., 2016; Schaeffer, 2017). In machine learning (ML), we want to replace the ground-truth map by a representation or approximation that is easier to store, apply, or simulate, such as a neural network. In this paper, we call these candidate models. A machine-learning pipeline may provide a countable list of finitely represented architectures and parameter settings (e.g., in `bfloat16` format); the theory does not range over all real-valued parameter choices. For proof generators, the set of candidate models could be the class of symbolic rule-based engines that take statements down to axioms. In this paper, candidate models can be any countable collection of maps, denoted by $\mathcal{M}_1, \mathcal{M}_2, \ldots$. We assume exact realizability, i.e., we assume that at least one listed candidate equals the ground-truth map. Note that this can also represent approximation rather than literal equality: if we only require accuracy up to a tolerance $\varepsilon$, we can partition the output space into bins of width $\varepsilon$ and treat outputs in the same bin as equivalent. Exact realizability of the resulting discretized map then corresponds to $\varepsilon$-accurate approximation in the original space.

Why is it that some data-driven inference problems are universally solvable while others are not? We find that the way in which the ground-truth map and candidate models are observed greatly affects whether identification, prediction, or verification is possible. The examples in Appendix A illustrate this point conditionally: they do not assume that current systems satisfy the countability, exact-realizability, noiselessness, or exhaustive-oracle assumptions of the theory. We will show that the way in which the ground-truth map

---

[1]The figure illustrates the three formal tasks only; it does not assert that present text-to-image systems satisfy the paper's oracle assumptions.

is observed (the type of training data) and the way in which candidate models are observed (the type of testing data) make some inference goals solvable and others not.

To this end, we define three main ways that maps are observed: (a) Passive observation, where one has no control over a map's inputs and has to accept whatever data is given, (b) Active observation, where one can decide what inputs to try, and (c) Membership-query, where one has an input-output pair and can check if the map can go between the two. We retain the shorter word "testing" in figures and tables. We prove that the way one observes the ground-truth and candidate maps directly influences whether map identification, map prediction, or map verification is universally solvable.

**Meaning of universal solvability.** A universally solvable task admits one deterministic learner that succeeds for every exactly realizable instance and every admissible, adversarially ordered exhaustive enumeration. Success is only in the limit: after a finite, instance-dependent time, every subsequent answer is correct, but the learner receives no certificate that this time has arrived and the definition imposes no uniform sample, query, or computational bound.

Since there are three inference goals (i.e., map identification, map prediction, and map verification), three ways to observe the ground-truth map (i.e., passive, active, and testing), and three ways to observe the candidate models (i.e., passive, active, and testing), there are $3 \times 3 \times 3 = 27$ types of problems one can investigate for each type of ground-truth map. For single-valued maps, i.e., maps for which every input is valid and gives a unique output, all 27 problems are universally solvable with an algorithm (see theorem 4). For finite-output maps, which can have finitely many possible outputs for one input or no output for an invalid input, we show that only 9-out-of-27 goals are universally solvable (see sections 3 to 5). For possibly-infinite-output maps, only 3-out-of-27 goals are solvable. We present concrete examples in Appendix A to demonstrate our framework while keeping the detailed practical discussion outside the introduction.

**Contribution and scope.** Our contribution is to determine, for all 81 combinations of inference goal, map type, and ways of observing the ground-truth and candidate maps, whether the task is universally solvable. We also explain which conclusions follow from existing language-learning results and which require arguments specific to our map setting. We do not present the 81 cells as 81 unrelated new theorems. Instead, we isolate (i) cells inherited by reduction from language identification, (ii) prompted-generation cells that adapt the critical-language idea of Kleinberg & Mullainathan (2024), (iii) standard membership-query comparisons, and (iv) arguments specific to the present operator formulation. The last group concerns arbitrary target-valid prompts rather than only robust prompts, separate access to the target and candidates, empty-fiber verification, and the loss of negative information when an active fiber changes from finite to infinite. The cell-by-cell accounting appears in Appendix B.

## 1.1 Related work

The work in this paper relies on arguments from language generation and set identification to make conclusions about map identification and map prediction. In set identification, there is an unknown countably infinite set $S_*$ (similar to our ground-truth map) and a sequence of countably infinite sets $S_1, S_2, \ldots$ (similar to our candidate models) for which $S_j = S_*$ for some $j$. The set identification problem is often thought of as a game between an adversary and an algorithm. In each round of the game, an element of $S_*$ is revealed to the algorithm (similar to a passive observation from the ground-truth map), and the algorithm may ask if an element $s$ is contained in a set $S_i$ for some $i$ (similar to an active observation of our candidate models). After each round, the algorithm should give its current best guess at an integer $j$ such that $S_j = S_*$.

In 1967, Gold showed that there does not exist an algorithm to universally solve set identification problems (Gold, 1967). This means that for any algorithm, there is a set identification problem for which the algorithm incorrectly guesses candidate sets for infinitely many rounds. If one thinks of a language as a countably infinite collection of valid words, then Gold essentially showed that one cannot learn a language by passively listening to it. We use Gold's ideas in one of our impossibility proofs by showing that all set identification problems are map identification problems. Angluin later characterized identifiable language families through finite tell-tales; see Angluin (1980); Jain et al. (1999). Accordingly, our passive-target identification lower bounds are reductions from this classical framework rather than independent rediscoveries.

Language generation is studied in Kleinberg & Mullainathan (2024). Language generation is not a set identification problem, though it has the same setup with a different goal. In language generation, one does not want to select an integer $j$ such that $S_j = S_*$. Instead, one wants to select an integer $j$ and an $s \in S_j$ such that $s \in S_*$. This is similar to our goal of map prediction. Remarkably, it was proved that language generation is solvable under Gold's observation model. If one thinks of a language as a countably infinite collection of valid words, then Kleinberg and Mullainathan essentially showed that one can parrot correct words of a language by passively listening to it. We use some of the ideas and techniques in Kleinberg & Mullainathan (2024) to show that some inference goals are solvable.

Compared to language identification and generation, which is modeled by a countably infinite set of valid strings, our setting is more intricate because it is defined by both the input set and the output(s) corresponding to an input. This gives rise to more scenarios to consider, depending on the cardinality of the output set (see Definition 1) and ways to observe the ground-truth map and candidate models (see Definition 2-4). At the level of represented objects, Equation (1) makes the correspondence exact. Our additional structure comes from two sources: retaining the fibers associated with individual prompts, and giving the learner different access approach to the target and candidate maps.

Recent work distinguishes prompted, uniform, non-uniform, and exhaustive generation, and studies optional feedback, breadth, hallucination, mode collapse, and stability (Raman et al., 2025; Charikar & Pabbaraju, 2025; Kalavasis et al., 2025b;a). Hallucination detection from positive data has also been related to identification, with negative examples changing the feasibility boundary (Karbasi et al., 2025). Noise, loss, membership feedback, and augmented observations lead to other hierarchies (Bai et al., 2026); those models provide information that is absent from our fixed access pair. Our prediction task asks only for one eventually valid witness, not novelty, breadth, representative generation, calibrated probabilities, or noise robustness.

Testing in our framework is the classical membership-query primitive, whereas active access is a chosen-input completion oracle. The broader exact-query literature also considers equivalence, subset, and related queries (Angluin, 1988). Our game differs from online mistake-bound learning because the learner receives no correctness feedback (Littlestone, 1988). It also differs from statistical model selection and hypothesis testing, which posit a sampling distribution and compare likelihoods, risks, or error probabilities (Vuong, 1989; Lehmann & Romano, 2005). Appendix B gives a more detailed provenance of all 81 cells.

## 1.2 Limitations and core assumptions

Our theoretical analysis makes a few idealized assumptions. First, we assume that the input and output spaces $\mathcal{X}$ and $\mathcal{Y}$ of the map $\mathcal{M}_*$ are nonempty and countable, and that the ground-truth map $\mathcal{M}_*$ lies within a known, countable class of candidate maps $\{\mathcal{M}_i\}$. These assumptions are not intended to capture the full richness of practical learning scenarios, where $\mathcal{X}$ is often continuous and the true mechanism may lie outside any prescribed model class, but they enable precise reasoning about what kinds of information make a task feasible in principle. By working in a countable setting, we separate the information-theoretic aspects of discovery from questions of optimization or approximation. We additionally assume exact realizability, noiseless observations, exhaustive (but possibly adversarially ordered) enumerations, and finitely many queries in each round. Countability of $\mathcal{X}$ and $\mathcal{Y}$ does not by itself make an unrestricted real-parameter family countable; the candidate list is additional prior information, and continuous scientific problems require a discretization or another countable restriction before the theorems apply.

Demonstrating that a particular inference goal is not universally solvable does not preclude progress. Our results only show that there is no universal algorithm to solve that type of inference, which means that for any algorithm there is at least one specific setup in that class for which the algorithm fails to accomplish the goal. One could potentially make progress by adding more problem-dependent knowledge to the algorithm or coming up with a different way to collect training or testing data. Similarly, if an inference task is universally solvable, our framework does not pinpoint exactly when a universal algorithm has succeeded at the task, just that with enough data it will eventually succeed. There is no uniform sample, query, or computational bound, and the learner receives no certificate that stabilization has occurred. Despite these limitations, the classification offers prescriptive value: it clarifies which types of observations are sufficient for successful inference. We hope it is a guide for data-driven discovery in the sciences.

### 1.3 Summary of our results and main takeaways

In section 6, we show that for single-valued maps where all the inputs are valid, map identification, map prediction, and map verification are all solvable for any type of observation. This is good news because many tasks in science are for maps that are single-valued.

In sections 3 to 5, we consider the case where the ground-truth map is finite-output, which means it is a set-valued map where each output is a set of finite cardinality. These maps can also occur for single-valued maps where some inputs are not valid (e.g., a chess engine that is given an invalid board position) and we want to regard the map as returning the empty set.

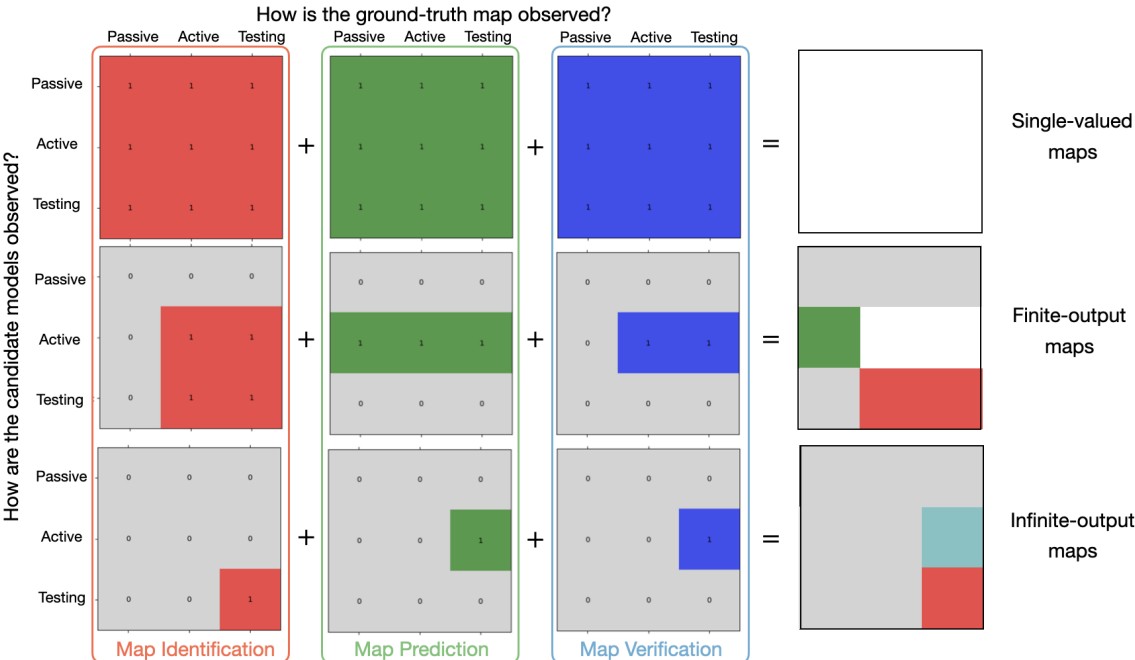

Figure 2: **Solvability landscape for data-driven inference across 81 scenarios.** Rows indicate the type of observation for candidate models (passive, active, testing) and columns indicate the type of observation for the ground-truth map, grouped by inference goal: map identification (red), prediction (green), and verification (blue). Separate blocks correspond to different ground-truth map structures: single-valued (top), finite-output (middle), and possibly-infinite-output (bottom). Colored cells mark scenarios that are universally solvable while gray cells indicate impossibility.

In the remainder of this paper, we present results about three types of ground-truth maps: (i) Single-valued, (ii) Finite-output, and (iii) Possibly-infinite-output. For each one we have 27 results, corresponding to the type of inference task and the types of observations. In section 2, we formally define a map and the three different ways to observe it. We study the feasibility of map identification, map prediction, and map verification given the various ways to observe the ground-truth map and candidate models in sections 3 to 5, respectively. In section 7, we conclude our analysis by studying set-valued maps for which the set contains infinitely many things for a given input.

## 2 How do we observe a ground-truth map and candidate models?

Ground-truth maps in science can be linear, nonlinear, single-valued, or set-valued. Here, we take a general point-of-view of a map and regard it as any function that takes objects from an input space $\mathcal{X}$ to objects in $\mathcal{Y}$ or subsets of $\mathcal{Y}$. Throughout our work, the input and output spaces of the candidate models match those of the ground-truth map.

**Standing assumptions.** The input space $\mathcal{X}$ and output space $\mathcal{Y}$ are nonempty countable sets, so each admits an enumeration. The ground-truth map $\mathcal{M}_*$ and every candidate model have domain $\mathcal{X}$ and take values in $\mathcal{P}(\mathcal{Y})$. The candidate models form a countable list $\mathcal{M}_1, \mathcal{M}_2, \ldots$, and there is an index $j_* \geq 1$ such that $\mathcal{M}_{j_*} = \mathcal{M}_*$; throughout, $j_*$ denotes the least such index. Thus, we work in an exactly realizable setting. We assume that all observations are noiseless and consistent with the corresponding map. Whenever an observation involves an enumeration, the enumeration is exhaustive and its order may be selected adversarially before the game is played. The ground-truth and candidate oracles have independently fixed enumeration orders, even when two maps are equal.

For single-valued maps, every input $f \in \mathcal{X}$ corresponds to a unique output element $g$ of $\mathcal{Y}$, i.e., $\mathcal{M}f = \{g\}$. For example, a uniquely solvable PDE is associated with a solution operator $\mathcal{M}_*$ that maps a forcing term $f \in \mathcal{X}$ to a unique solution $g \in \mathcal{Y}$. In section 6, we show that map identification, map prediction, and map verification are all solvable when the ground-truth map and candidate models are single-valued, regardless of the type of observations.

However, there are also set-valued maps. There are two main reasons for set-valued maps: (1) The ground-truth map is set-valued (e.g., there are many valid continuation tokens in AI-generated text), or (2) The underlying unknown system has no output for some $f \in \mathcal{X}$ (e.g., there is no valid proof of a false statement). We define our ground-truth map as returning the empty set if the underlying system does not give an output for some $f \in \mathcal{X}$. To write single-valued and set-valued maps in a consistent notation, we regard both as functions from $\mathcal{X}$ to $\mathcal{P}(\mathcal{Y})$, where $\mathcal{P}(\mathcal{Y})$ is the power set of $\mathcal{Y}$. For single-valued maps, the outputs are always singleton sets.

**Definition 1.** *Let $\mathcal{X}$ and $\mathcal{Y}$ be nonempty countable sets. A map $\mathcal{M} : \mathcal{X} \to \mathcal{P}(\mathcal{Y})$ is a function that takes every input element $f \in \mathcal{X}$ to a subset of $\mathcal{Y}$, i.e., $\mathcal{M}f \subseteq \mathcal{Y}$. We assume that every map has at least one valid input-output pair, i.e., there exist $f \in \mathcal{X}$ and $g \in \mathcal{Y}$ such that $g \in \mathcal{M}f$. We call $\mathcal{M}$ finite-output if $\mathcal{M}f$ is finite for every $f \in \mathcal{X}$, and single-valued if $\mathcal{M}f$ is a singleton for every $f \in \mathcal{X}$.*

We denote the ground-truth map by $\mathcal{M}_*$, which is a function from $\mathcal{X}$ to $\mathcal{P}(\mathcal{Y})$. If $f$ is not a valid input to a map, then we consider it as outputting the empty set. Moreover, there is a list of candidate models $\mathcal{M}_1, \mathcal{M}_2, \ldots$ with $\mathcal{M}_i : \mathcal{X} \to \mathcal{P}(\mathcal{Y})$. At least one candidate is exactly equal to $\mathcal{M}_*$, but we do not know a priori which one it is; candidates need not be distinct. To solve our inference goal of map identification, map prediction, or map verification, we collect observations from the ground-truth map and the candidate models. At no point are we told whether our current selection or output is correct; however, we are allowed to collect more observations in later rounds.

The types of observations that we make on the ground-truth map and the candidate models are crucial. We now formally define passive, active, and testing observations.

**Definition 2** (Passive). *Let $\mathcal{M} : \mathcal{X} \to \mathcal{P}(\mathcal{Y})$ and define its set of valid input-output pairs by $\mathcal{A}_{\mathcal{M}} = \{(f, g) \in \mathcal{X} \times \mathcal{Y} : g \in \mathcal{M}f\}$. The adversary fixes an exhaustive enumeration of $\mathcal{A}_{\mathcal{M}}$. If $\mathcal{A}_{\mathcal{M}}$ is finite with cardinality $n \geq 1$, write the enumeration as $(f_1, g_1), \ldots, (f_n, g_n)$ and repeat it cyclically; the $k$th passive observation is $(f_\ell, g_\ell)$, where $\ell = ((k-1) \bmod n) + 1$. If $\mathcal{A}_{\mathcal{M}}$ is countably infinite, write its enumeration as $(f_1, g_1), (f_2, g_2), \ldots$; the $k$th passive observation is $(f_k, g_k)$.*

In a passive observation, one has no control over the input or the output. In many other situations, such as conducting an experiment, we can specify an input and see the output. We call this an active observation.

**Definition 3** (Active). *Let $\mathcal{M} : \mathcal{X} \to \mathcal{P}(\mathcal{Y})$ and fix an input $f \in \mathcal{X}$. If $\mathcal{M}f = \emptyset$, every active observation with input $f$ returns the distinguished symbol $\perp$, which informs the algorithm that the output set is empty. If $\mathcal{M}f$ is finite and nonempty with cardinality $n_f$, the adversary fixes an ordering $g_1(f), \ldots, g_{n_f}(f)$ of its elements and repeats it cyclically; the $k$th active observation with input $f$ returns $g_\ell(f)$, where $\ell = ((k-1) \bmod n_f) + 1$. If $\mathcal{M}f$ is countably infinite, the adversary fixes an exhaustive enumeration $g_1(f), g_2(f), \ldots$, and the $k$th active observation with input $f$ returns $g_k(f)$. Here, $k$ counts only the active observations made of the same map with the same input $f$.*

Hence, a key difference between a passive and an active observation is that in an active observation, the input $f \in \mathcal{X}$ is chosen; in a passive one, it is not. If $\mathcal{M}f = \emptyset$, an active observation returns $\perp$. If the output set is finite and nonempty, repeated observations eventually return every possible output and then repeat

in a loop, so the first repetition reveals the whole fiber. If there are countably infinitely many outputs, the observations never repeat and do not certify that an unobserved output is absent.

**Definition 4** (Testing (membership query))**.** *Let $\mathcal{M} : \mathcal{X} \to \mathcal{P}(\mathcal{Y})$. A test of $\mathcal{M}$ with an input-output pair $(f, g) \in \mathcal{X} \times \mathcal{Y}$ returns the Boolean value 1 if $g \in \mathcal{M}f$ and 0 otherwise.*

Therefore, for testing, one must propose both an input $f$ and output $g$, and a test is an observation of a single Boolean value revealing if $g \in \mathcal{M}f$. Thus, testing is exactly a membership query to the set of valid input-output pairs. Unlike an active observation, it can certify that a particular proposed $g$ is absent, but finitely many negative tests need not certify that the whole fiber is empty. A test happens most often in learning with oracles. For example, given an encryption algorithm, it is often much easier to verify if a password is correct than to generate one. It is also much easier to verify the validity of a proof of a challenging theorem than to generate one. This makes testing observations more practical for certain inference goals (see section 7).

Classical language learning can be viewed as a special case of our framework in two useful ways. First, let the input space contain a single element, $\mathcal{X} = \{f_0\}$, and let $\mathcal{Y}$ contain all possible strings. For any language $L \subseteq \mathcal{Y}$, define

$$\mathcal{M}_L f_0 = L. \tag{1}$$

Under this representation, passively observing the map is the same as observing valid strings from $L$, identifying the map is the same as identifying $L$, and producing an output from $\mathcal{M}_L f_0$ is the same as generating a valid string.

Alternatively, the same language dataset can be viewed as a collection of valid inputs. Let $\mathcal{X}$ contain all possible strings, let $\mathcal{Y} = \{1\}$, and define $\mathcal{M}_L f = \{1\}$ if $f \in L$ and $\mathcal{M}_L f = \emptyset$ otherwise. Passive observations then reveal positive examples from $L$, while testing whether $1 \in \mathcal{M}_L f$ is exactly a language-membership query. This second representation is used to reduce classical language identification to our map-identification and verification problems, as explained in Appendix B.

Our general setting goes beyond both special cases by allowing many inputs, each of which may have several valid outputs. For a given input $f$, we call the set $\mathcal{M}f$ its fiber. Organizing valid outputs by their inputs is what allows us to study prompted prediction and verification, and it is also where some of our arguments differ from standard language-learning results.

## 3 When can we identify the ground-truth map for finite-output maps?

In this section, we study map identification for finite-output maps. There is a ground-truth map $\mathcal{M}_* : \mathcal{X} \to \mathcal{P}(\mathcal{Y})$ and a set of candidate models $\mathcal{M}_1, \mathcal{M}_2, \ldots$, with $\mathcal{M}_i : \mathcal{X} \to \mathcal{P}(\mathcal{Y})$, and one wishes to identify an index $j$ for which $\mathcal{M}_j = \mathcal{M}_*$. Throughout this section, maps are finite-output, meaning that $\mathcal{M}f$ is a set of finite cardinality for any $f \in \mathcal{X}$. The inference task can be formulated as a game played between an algorithm and an adversary (see Figure 3).

**The map identification game.** For every map $\mathcal{M}$ (which can be the ground-truth map $\mathcal{M}_*$ or a candidate model $\mathcal{M}_j$ for some $j \geq 1$), the adversary fixes the passive enumeration of $\mathcal{A}_\mathcal{M}$ and, for every $f \in \mathcal{X}$, the active enumeration of $\mathcal{M}f$ described in Definitions 2 and 3. In each round, the following happens:

1. **Observations from the ground-truth map.** The algorithm collects observations from $\mathcal{M}_*$, which may be passive, active, or testing. For a passive or active observation, it receives the next item in the adversary's enumeration (see Definitions 2 and 3).
2. **Observations from a candidate model.** The algorithm collects observations from the candidate models, which could be passive, active, or testing. When the algorithm makes a passive or active observation, it gets the next observation in the adversary's enumeration (see Definition 2 and 3).
3. **Identification output.** The algorithm outputs a candidate index $j$.

We say that the algorithm wins the map identification game for $\mathcal{M}_*$ if, after a finite number of rounds, it always selects a candidate model with $\mathcal{M}_j = \mathcal{M}_*$ in the "identification output" step. So, even if more

rounds are played, the algorithm sticks with a correct candidate model after some point. (This prevents the algorithm from getting lucky in one round.) Importantly, at no point in the game is the algorithm told that it has selected a correct candidate model; it just collects more observations in subsequent rounds and keeps its choice.

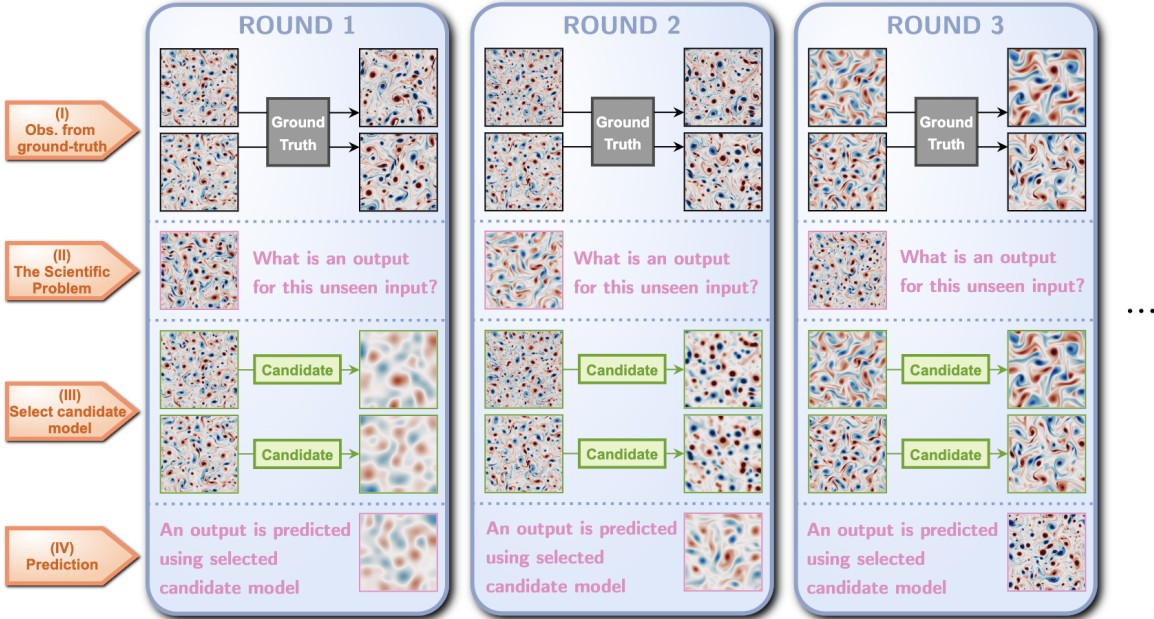

Figure 3: **Workflow of our learning game.** We cast data-driven inference as a game between a learner and an adversary. In each round, the learner gathers observations from the ground-truth map, receives a question from the adversary, and selects a candidate model to answer the question. For prediction, for example, the question may involve forecasting the next state in a simulated physical system. Over successive rounds, the learner refines the selection of the candidate model.

Formally, the algorithm can be modeled as an oracle machine, a Turing machine equipped with access to oracles that provide information about the ground-truth map and candidate models. The nature of oracle access depends on the type of observation (passive, active, or testing), and the order of an oracle's output may be arbitrary. A candidate query specifies its index; an active query also specifies $f$, and a testing query specifies $(f, g)$. The machine may make any finite number of queries in a round, but it must eventually return an answer. That is, the machine loses the game if it does not halt in any round of the game. In this formulation, the input tape remains empty across rounds because the identification task is fixed.

**Definition 5** (Map identification is universally solvable.)**.** *We say map identification is universally solvable if there is a deterministic algorithm that always wins the map identification game.*

If map identification is universally solvable, then there is a mechanism for identifying a candidate model equal to $\mathcal{M}_*$ for any instantiation of the problem. The adversary's role is to select enumerations that make this as difficult as possible. The way we observe maps is therefore crucial in determining whether map identification is solvable.

**Theorem 1.** *Under the standing assumptions, map identification for finite-output maps is universally solvable if and only if both the ground-truth map and candidate models are observed actively or via testing.*

**Proof idea.** For finite-output maps, active observations can mimic tests: repeated queries either return $\perp$ or reveal the whole finite fiber when the outputs begin to repeat. We can therefore reduce the positive cases to testing both maps. The algorithm tries the candidates in order and compares each one with the target on an enumeration of $\mathcal{X} \times \mathcal{Y}$. Every wrong candidate eventually disagrees on some pair and is discarded, while a correct one is never discarded. For the negative cases, a passive stream only reveals positive pairs. A

missing pair can always be postponed beyond the finite transcript already seen, which gives the Gold-style target obstruction and the corresponding passive-candidate diagonal construction.

See Appendix C for the proof. Theorem 1 gives a complete characterization: for an algorithm to always win the map identification game, neither the ground-truth map nor the candidate models can be passively observed.

## 4    When can we predict the ground-truth map for finite-output maps?

For some unknown systems, our objective is not to identify the ground-truth map itself, but rather to correctly predict its output for any given input. In particular, one does not accomplish map identification to succeed at map prediction. For example, in solving PDEs with neural operators, one need not recover the full differential operator governing the system; instead, it is sufficient to approximate the mapping from boundary conditions to solutions (Rudy et al., 2017). Similarly, in fluid flow prediction, a learned model can accurately forecast flow evolution without explicitly recovering the Navier–Stokes operator, as long as it captures the correct input-output relationships.

Here, we study map prediction for finite-output maps with the same setup as in section 3. This task can also be formulated as a game played between an algorithm and an adversary.

**The map prediction game.**    For every map $\mathcal{M}$, the adversary fixes the passive and active enumerations described in Definitions 2 and 3. Then, in each round, the following happens:

1. **Observation from the ground-truth map.** The algorithm collects observations from $\mathcal{M}_*$, which may be passive, active, or testing. For a passive or active observation, it receives the next item in the adversary's enumeration (see Definitions 2 and 3).
2. **Adversary's selected input.** The adversary selects an $f \in \mathcal{X}$ for which $\mathcal{M}_* f \neq \emptyset$.
3. **Observations from a candidate model.** The algorithm collects observations from the candidate models, which could be passive, active, or testing. When the algorithm makes a passive or active observation, it gets the next observation in the adversary's enumeration (see Definition 2 and 3).
4. **Prediction output.** The algorithm outputs an index $j$ and an element $g \in \mathcal{M}_j f$.

The map prediction game differs from the identification game in that the adversary chooses a new $f$ each round for which the algorithm must predict a $g \in \mathcal{M}_* f$. We say that the algorithm wins if, after a finite number of rounds, it always correctly selects such a $g$. The input $f$ can change in every round. The pair $(j, g)$ is only a provisional round output; it is not an optimizer under a loss or score. Importantly, the algorithm is never told whether its prediction is correct.

**Definition 6** (Universal solvability of map prediction.)**.** *We say map prediction is universally solvable if there exists a deterministic algorithm that always wins the map prediction game.*

On the one hand, the map prediction game looks easier than the map identification game because the algorithm only has to correctly predict a valid output for one input per round. On the other hand, it is harder because the adversary can change the input $f$ every round. In particular, the map prediction game can be unsolvable when the map identification game is solvable and vice versa.

**Theorem 2.** *Under the standing assumptions, for finite-output maps, the map prediction game is universally solvable if and only if the candidate models are observed actively.*

**Proof idea. Why active candidate observations are sufficient:**    Suppose that the candidate models can be actively observed. When the adversary asks for a prediction at an input $f_t$, repeated active observations reveal the complete finite output set $\mathcal{M}_j f_t$ of any candidate $\mathcal{M}_j$: they return $\perp$ if the set is empty, and otherwise eventually repeat after showing every output. The algorithm also keeps all ground-truth input-output pairs observed so far and rules out candidates that disagree with any of them. Among the remaining candidates, it chooses one whose output set at $f_t$ is contained in the output sets of all earlier eligible candidates. Eventually, every candidate appearing before the first correct candidate is either ruled

out or contains all outputs of the ground-truth map. From that point onward, the chosen candidate's output set at $f_t$ is contained in $\mathcal{M}_* f_t$, so any output returned from it is correct. This argument only requires passive observations of the ground truth, which can be obtained under any of the three ground-truth observation modes given all fibers are finite.

**Why testing candidate models is not sufficient:** The difficulty is that testing can check only one proposed output at a time. Consider a new input $f$ and suppose that the algorithm tests a candidate on finitely many outputs $g_1, \ldots, g_m$, receiving a negative answer each time. These answers do not tell the algorithm whether the candidate has no valid output at $f$ or has one valid output $g$ that has not yet been tested. In the first case, the algorithm must use another candidate; in the second case, the candidate being tested may itself be correct. Because the algorithm must stop after finitely many tests, it cannot always distinguish these two cases. Active candidate observations provide exactly the missing information: they return either a valid output or $\perp$. The proof in Appendix D repeats this ambiguity on a new input in each round and constructs a fixed problem on which the algorithm makes infinitely many incorrect predictions.

See Appendix D for the proof. We see from Theorem 2 that map prediction is universally solvable when the ground-truth map is passively observed and the candidate models are actively observed, but map identification is not. Similarly, map identification is universally solvable when the candidate models are observed via testing and the ground-truth map is actively observed, but map prediction is not: pairwise tests do not certify that a fresh candidate fiber is nonempty.

## 5 When can we verify an input for finite-output maps?

There are many unknown systems for which the given input to the corresponding ground-truth map may not be valid. For example, when we ask an automated proof generator to prove a false statement, the set of valid proofs for that statement is empty. Or, when we ask a large language model to produce the social security number of an individual, we want the map not to reveal its knowledge (even if it knows it). In that case, we want to learn a ground-truth map with input verification.

In this section, we study map verification for finite-output maps with the same setup as in sections 3 and 4. The inference task can again be formulated as a game played between an algorithm and an adversary.

**The map verification game.** For every map $\mathcal{M}$, the adversary fixes the passive and active enumerations described in Definitions 2 and 3. Then, in each round, the following happens:

1. **Observations from the ground-truth map.** The algorithm collects observations from $\mathcal{M}_*$, which may be passive, active, or testing. For a passive or active observation, it receives the next item in the adversary's enumeration (see Definitions 2 and 3).
2. **Adversary's selected input.** The adversary selects an arbitrary $f \in \mathcal{X}$ (possibly with $\mathcal{M}_* f = \emptyset$).
3. **Observations from a candidate model.** The algorithm collects observations from the candidate models, which could be passive, active, or testing. When the algorithm makes a passive or active observation, it gets the next observation in the adversary's enumeration (see Definitions 2 and 3).
4. **Verification output.** The algorithm outputs a Boolean indicating whether $\mathcal{M}_* f$ is empty. If it outputs "nonempty," it also outputs an index $j$ and an element $g \in \mathcal{M}_j f$.

We say that the algorithm wins the map verification game for $\mathcal{M}_*$ if, after a finite number of rounds, it always returns the right Boolean value and, whenever the input is valid, a $g \in \mathcal{M}_* f$. The input can change in each round. At no point is the algorithm told that it has selected the correct Boolean value or a valid output.

**Definition 7.** *We say map verification is universally solvable if there is a deterministic algorithm that always wins the map verification game.*

Verification cannot be reduced to prediction merely by adjoining a symbol $g_0$ to every fiber: a predictor may always return $g_0$ and need not reveal whether an original output exists. If $g_0$ is instead placed only in

empty fibers, constructing the transformed oracle already requires deciding emptiness, which is precisely the extra information verification asks for. Thus verification genuinely adds an emptiness decision to witness prediction.

**Theorem 3.** *Under the standing assumptions, for finite-output maps, the map verification game is universally solvable if and only if the ground-truth map is observed actively or via testing and the candidate models are observed actively.*

**Proof idea.** For the positive direction, run the identification algorithm in the background. Once it has settled on a correct candidate, one active query to that candidate returns $\perp$ exactly for an empty fiber and otherwise gives a valid witness. For the negative direction, verification contains prediction on nonempty inputs, so Theorem 2 rules out passive or testing candidate access. The remaining passive-target case reduces Gold's set-identification problem to verification: stabilized Boolean answers on longer and longer blocks would reveal enough membership information to identify the target set.

See Appendix E for the proof. Comparing Theorem 3 with Theorem 2, the only difference arises when the ground-truth map is observed passively and the candidate models are observed actively. In the idealized text-generation mapping of Appendix A, this is the formal separation between prediction and verification, although it is not a claim that practical AI alignment is impossible.

## 6 All inference tasks are solvable for single-valued maps

Single-valued maps are those in which each input corresponds to a unique valid output, i.e., $\mathcal{M}f$ is a singleton for every ground-truth and candidate map. Many unknown systems involve such maps, and in this setting we can show that all three inference goals are universally solvable, regardless of how the maps are observed.

**Theorem 4.** *Under the standing assumptions, when the ground-truth map and every candidate model are single-valued, map identification, map prediction, and map verification are universally solvable no matter how the ground-truth map and candidate models are observed.*

**Proof idea.** Every access mode eventually reveals the unique output on every input: passive access enumerates the graph, active access lets us query the input, and testing access can enumerate possible outputs until the unique positive answer is found. If a candidate differs from the target, their positive observations eventually reveal different outputs on the same input, so the wrong candidate can be discarded without any negative data. After identification, prediction returns the unique output and verification always reports that the fiber is nonempty.

See Appendix F for the proof. Theorem 4 shows that all inference goals are universally solvable for single-output maps regardless of the type of observations.

## 7 Necessity of negative data: infinite-output maps

So far, we have assumed that all maps are single-output or finite-output; that is, $\mathcal{M}f \subseteq \mathcal{Y}$ is finite for every $f \in \mathcal{X}$. In practice, however, there are many situations in which the number of outputs is infinite. For example, in automated theorem proving, a single statement may have infinitely many valid proofs. After a countable representation is fixed, a quantized initial condition in a stochastic forecasting model may likewise admit countably many encoded future trajectories. Continuous trajectories and probability distributions remain outside our exact framework unless they are separately discretized.

A key difference between finite-output and infinite-output maps lies in the information obtainable through active observations. For finite-output maps, sufficiently many active observations allow one to recover $\mathcal{M}f$ exactly: after enough queries, the outputs repeat, revealing both what is in $\mathcal{M}f$ and what is not. This "negative information" is a standard concept in theoretical computer science and plays a crucial role in identification (Gold, 1967; Angluin, 1980). In contrast, for infinite-output maps, active observation alone cannot reveal whether a given element is not in $\mathcal{M}f$. Consequently, an active observer cannot, in general, perform a definitive membership test for $(f, g)$ using active observations alone.

**Definition 8.** *A possibly-infinite-output map identification, map prediction, or map verification game is the corresponding game from Sections 3 to 5, respectively, in which some fibers of the ground-truth map or candidate models may be countably infinite. This class contains the finite-output maps as a subclass.*

Not surprisingly, the lack of negative data severely limits what can be achieved with infinite-output maps. We will show that only a few special cases admit a universal algorithm.

Here is the main intuition. When a map can have infinitely many outputs, observing more outputs never tells us whether some unseen output also exists. To identify the entire map, we therefore need to test individual input-output pairs for both the ground truth and the candidate models. Prediction and verification require less information. An active observation of a candidate either provides a possible output or reports that no output exists, and tests of the ground truth can eventually reveal whether this answer is wrong. Whenever a candidate gives a wrong answer, the algorithm eventually rules it out and moves to the next one. Since the correct candidate appears at a finite position in the list, the algorithm eventually reaches a candidate that always answers correctly.

**Theorem 5.** *Under the standing assumptions, the following statements about possibly-infinite-output problems hold:*

1. *There exists an algorithm to universally solve the possibly-infinite-output map identification game given any ground-truth and candidates if and only if it performs tests of the ground-truth and of the candidate models.*
2. *There exists an algorithm to universally solve the possibly-infinite-output map prediction game given any ground-truth and candidates if and only if it performs tests of the ground-truth and makes active observations of the candidate models.*
3. *There exists an algorithm to universally solve the possibly-infinite-output map verification game given any ground-truth and candidates if and only if it performs tests of the ground-truth and makes active observations of the candidate models.*

**Proof idea.** For identification, testing both graphs lets us search for a disagreement and discard every wrong candidate. For prediction and verification, we try candidates in stages. A candidate either gives a witness, which is tested against the target, or declares the fiber empty, in which case we test more and more possible target outputs. A wrong answer is eventually exposed; a candidate that is never exposed is already answering correctly. The negative results combine the finite-output lower bounds, the Gold-style and passive-candidate reductions, and a diagonal argument in which every finite active transcript can still be extended in two incompatible ways on a fresh prompt.

See Appendix G for the proof. To illustrate Theorem 5, consider the personalized spam-filtering example in Appendix A. The theorem says that exact identification in that formal model requires testing the ground truth: seeing only emails labeled as spam does not reveal that an unobserved email is non-spam. This does not rule out useful statistical filters under a distribution; it identifies the negative information required for exact universal identification.

## 8 Conclusion

The results in this paper highlight how the universal solvability of identification, prediction, and verification tasks depends critically on both the nature of the underlying maps (e.g., single-valued vs. infinite-output) and the type of observational access available to the ground-truth map and candidate models. We illustrate how theoretical results translate into practical considerations for the design and evaluation of scientific and AI systems. Our results underscore that, while universal solvability can be achieved for single-valued maps under any observation regime, infinite-output settings require more stringent conditions, such as access to testing oracles. Our findings provide a unified perspective that connects information-theoretic constraints with methodological choices within the paper's idealized assumptions.

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

# A    Applications of our framework

The examples in this appendix illustrate how a practical problem might be mapped to the formal games. Each conclusion is conditional on the standing assumptions: countable spaces and candidate classes, exact realizability, noiseless observations, exhaustive oracle enumerations, and eventual rather than resource-bounded correctness. The mappings do not establish that a current system satisfies these assumptions, nor do formal impossibility results rule out guarantees under additional domain structure, approximation, distributions, or feedback.

## A.1    AI text generation

One of the most popular map prediction tasks in recent years is large language models (LLMs) for text generation (Vaswani et al., 2017; Lewis et al., 2020; Achiam et al., 2023; Gu & Dao, 2023; Yu & Erichson, 2025). For our setup, suppose that there is a ground-truth map $\mathcal{M}_*$ that takes in a sequence of tokens and outputs a finite set of tokens that are valid continuations of the sequence in human text. The length of the sequence is referred to as the context window and, in practice, an LLM selects a token from the set of possible continuation tokens. The candidate family is a known countable list of finitely represented language models, and one candidate is assumed to equal the target map. We regard the ground-truth map as being observed passively through human text available on the internet. The candidate models are observed actively because, given any sequence of input tokens and a candidate LLM, we can run the model to obtain a next token. Under the remaining standing assumptions, Theorem 2 makes this access pattern universally solvable for map prediction. With the same passive target access, Theorem 3 does not give universal verification because positive text alone cannot certify that a requested fiber is empty. This formal separation does not show that present LLM training universally succeeds or that practical alignment is impossible, where validity may be noisy, context dependent, non-realizable, or supported by richer human feedback (Hendrycks et al., 2020).

## A.2    Who is speaking?

Turing has a famous imitation test, where participants have a five minute conversation with a human or a machine and at the end the participant has to guess if they were speaking to a human or a machine (Turing, 2009). A machine passes the imitation test if it fools enough participants that it is human. Here, we imagine that we are trying to develop an algorithm to determine who we are speaking with. Our ground-truth map takes in a conversational prompt and outputs a set of reasonable replies. The ground-truth map could be a human texter or an AI chatbot. The candidate models are all chatbots and human texters under consideration. The inference task here is map identification because we want to determine who is speaking. If the algorithm only gets to watch conversations, then we are passive observers of the ground-truth map. If the algorithm is actively engaged in the conversation and can also engage with any of the candidate models, then both sides are actively observed. Under exact realizability and finite fibers, Theorem 1 rules out a universal guarantee for the former access pattern and permits one for the latter. This is an in-limit statement for the stipulated candidate family, not a finite-conversation test for identifying real speakers.

## A.3    Designing proteins for biological function

David Baker's lab designs proteins to perform a particular biological function (Baek et al., 2021; Hedrick, 2009). In this setting, we regard the ground-truth map $\mathcal{M}_*$ as an operator that takes a biological function to the set of proteins that achieve that function. We take the candidate models to be a prescribed countable collection of finitely represented generative models, such as configurations in the RFdiffusion framework (Goodfellow et al., 2020; Kingma & Welling, 2013; Song et al., 2021; Huang et al., 2021; Song et al., 2020; Karras et al., 2022; Lipman et al., 2023; Lim et al., 2025; Watson et al., 2023). Our task is to select a candidate model that correctly outputs proteins for a given biological function, which is a map prediction task. The ground-truth map is observed passively with a database of known proteins and their biological functions. However, the candidate models are easy to evaluate for any specific input, making us active observers of them. If the relevant fibers are finite and one candidate exactly equals the target, Theorem 2 gives

the corresponding in-limit prediction result. Real protein design involves continuous representations, noisy assays, model misspecification, and resource constraints, so this is not a practical convergence guarantee.

### A.4    Personalized spam filters

Personalized spam filtering, where the classifier adapts to each user's definition of spam, is well-studied in the literature (Cormack & Lynam, 2007). Imagine that a ground-truth map takes a user as an input and maps it to all emails considered spam by that user. The candidate models could be classifiers that take in a user's profile and an email and classify it as spam or not. The goal is to identify which candidate model is the ground truth so that we can use it as a personalized spam filter. We imagine observing the ground-truth map by keeping track of the user's inbox and getting the user to flag spam emails. By the nature of candidate classifiers, we perform tests on them. One key property is that, given a user, there may be infinitely many emails considered spam by that user. For exact identification in this formal model, Theorem 5 requires testing access to the ground truth; positive or active enumeration alone does not certify that an unobserved email is non-spam. This does not imply that useful statistical spam filters are impossible, since they generally seek approximate performance under a distribution rather than exact equality on every possible email.

### A.5    Automatic theorem proving

Consider a formal logic system with a fixed set of axioms. The ground-truth map $\mathcal{M}_*$ takes a statement $P \in \mathcal{X}$ to the set $\mathcal{M}_* P$ of all valid proofs of $P$. When $P$ is provable, this set may be infinite; when $P$ is unprovable in the system, it is empty. Automatic theorem proving seeks to produce a valid proof when one exists and to correctly reject an input when none exists, framing the task as map verification. Modern AI theorem provers can be treated as candidate models; since they can be run on arbitrary inputs, we are active observers of the candidates. Within the possibly-infinite-output abstraction, Theorem 5 rules out universal verification from only a passive stream of provable statements and valid proofs. Its positive access pattern also requires the ability to test a purported proof against the formal logic (Harrison, 2009).

### A.6    Proof validation

Here, the ground-truth map is again defined by the logic system, but a candidate model takes as input a statement and a proposed proof, returning whether it deems the proof valid. Each candidate model thus maps a statement to the set of all proofs it accepts. We act as testers of these models. Proof validation does not require computing the ground-truth map explicitly; it can be represented by identifying a candidate whose behavior matches the formal system (Barbosa et al., 2023). For an exact universal guarantee in this model, Theorem 5 requires direct tests of $\mathcal{M}_*$; examples of valid proofs alone do not supply the negative information needed to compare arbitrary proposed proofs.

### A.7    PDE learning

In PDE learning, the ground-truth map is the underlying PDE solution operator that takes a forcing term to its solution. Candidate models include DeepONets (Lu et al., 2019), Fourier Neural Operators (Li et al., 2020), and physics-informed neural networks (Cai et al., 2021; Cuomo et al., 2022). The task may be map prediction (computing the solution for a given forcing term) or map identification (recovering the PDE form). Since the solution to a well-posed PDE is unique, the output set for each input is a singleton. The ground-truth map may be observed passively or actively, depending on whether we can choose the forcing term, while candidates are actively observed through simulation. The exact-realizability assumption does not require a candidate to reproduce the continuum solution operator perfectly. Instead, fix an accuracy level $\varepsilon$, choose a norm on the solution space, and divide the discretized outputs into bins whose diameter is at most $\varepsilon$. The output of the ground-truth map is then the bin containing the true solution. If a candidate always predicts the same bin as the ground truth, it exactly realizes this discretized map, while its prediction differs from the true solution by at most $\varepsilon$. We must also restrict the forcing terms to a prescribed countable set and the candidates to a countable list of finitely described configurations, such as networks with finite-precision parameters. Under these conditions, Theorem 4 gives a positive in-limit result for PDE learning at any fixed

numerical resolution. The theorem does not provide a sample-complexity bound or describe what happens as $\varepsilon \to 0$, but this approximation perspective makes the framework applicable beyond literal equality with the continuum solution operator.

### A.8 AI chess engines

Here, the ground-truth map assigns to a board state a set of good moves, as in AlphaZero (Silver et al., 2018). Candidate models include the AI algorithms under consideration, each defined by an architecture and parameter set. For the theory, these candidates are restricted to a prescribed countable list of finitely represented engine configurations. If the goal is to output only the best move, the problem reduces to a single-output setting. In many applications, however, we require a set of good moves, making the problem finite-output. The output set can also be empty for a terminal or invalid board state. In this general form, the task is map verification: given a board state, determine whether a good move exists and, if so, produce one. Candidates are actively observed, but the ground truth might be observed only passively from historical expert games. Under the standing assumptions, Theorem 3 rules out a universal guarantee for that exact access pattern; active or testing target access gives the theorem's positive pattern. This conditional result does not assess statistical performance from historical games.

# B Result provenance and relation to language learning

## B.1 Exact graph-language correspondence

The graph-language encoding is a one-to-one representation of maps: equality of maps is exactly equality of their graph languages. Passive map observations are positive texts for these languages, and testing observations are membership queries. Thus, map identification under passive target access contains classical language identification in the limit as the singleton-output special case $\mathcal{Y} = \{1\}$, with $\mathcal{M}_i(f) = \{1\}$ exactly when $f \in S_i$. The passive-target lower bounds in Theorems 1 and 3, and the active-target/testing-candidate identification lower bound in Theorem 5, use this reduction from the Gold–Anglin framework (Gold, 1967; Anglin, 1980).

## B.2 Generation, prompting, breadth, and feedback

Kleinberg and Mullainathan prove that, from a positive enumeration of any target in a countable language family, a learner can eventually generate unseen target strings (Kleinberg & Mullainathan, 2024). Their prompted theorem with ordinary membership queries assumes robust prompts, meaning every candidate has arbitrarily many completions; their result for arbitrary nontrivial prompts uses stronger regular-subset queries. Our prediction game removes the unseen-string requirement but permits every prompt with a nonempty target fiber, even when all early candidates have empty fibers there. The positive finite-output algorithm in Theorem 2 adapts their critical-language selection principle, with active candidate access supplying a completion or an empty-fiber certificate for the current prompt.

Subsequent work distinguishes uniform, non-uniform, and prompted generatability (Raman et al., 2025); exhaustive generation and generation with optional feedback (Charikar & Pabbaraju, 2025); tradeoffs between consistent generation, breadth, hallucination, and mode collapse (Kalavasis et al., 2025b); and the interaction between breadth and generator stability (Kalavasis et al., 2025a). Hallucination detection from positive data can be as hard as identification, whereas labeled negative examples can make it possible (Karbasi et al., 2025). Noise, loss, membership feedback, and augmented observations yield still different hierarchies (Bai et al., 2026); those models provide labels, feedback, or other signals absent from our fixed access pair. The present task asks only for eventual one-witness validity. It does not require an unseen output, exhaustive or representative coverage, calibrated probabilities, distributional correctness, a convergence-rate bound, or noise robustness.

## B.3 Query, statistical, and online-learning models

The broader exact-learning literature studies membership, equivalence, subset, superset, disjointness, and exhaustiveness queries (Anglin, 1988). Our testing oracle is precisely a membership query to $L_{\mathcal{M}}$. Active access is different: it returns a completion for a chosen prompt and certifies an empty fiber, but for an infinite fiber it cannot certify that a particular unobserved completion is absent. The finite-fiber simulation in Part I of Appendix C is therefore the bridge from active access to membership queries, and its failure for infinite fibers drives several separations. Classical model selection and hypothesis testing use probabilistic sampling models and finite-sample likelihood, risk, or error criteria (Vuong, 1989; Lehmann & Romano, 2005); our games have no sampling distribution or statistical loss. Online mistake-bound learning supplies correctness feedback after predictions and evaluates the number of mistakes (Littlestone, 1988); our games supply no such feedback and require only eventual correctness, with no uniform sample, query, computation, or mistake bound.

## B.4 Where the 81 results come from

The 81 cases arise from three inference tasks, three ways of observing the ground-truth map, three ways of observing the candidate models, and three types of output sets. Table 2 summarizes all of these cases. In each row, the access pair $(T, C)$ describes how the target and candidates are observed, respectively. We write P for passive observations, A for active observations, and Q for testing, or membership queries. Each

cell contains the code of the main argument prototype used in that case. Table 1 explains the codes and states whether each prototype establishes a universal algorithm or an impossibility result.

Table 1: Argument prototypes used in the 81 cases. The row colors indicate how each argument is related to earlier work.

| Code | Role | Argument prototype and relationship to earlier work | Complete argument |
|------|------|------|------|
| **Arguments taken from or adapted from earlier work** | | | |
| LI | Impossibility | Language-identification reduction from positive examples, following Gold and Angluin (Gold, 1967; Angluin, 1980). | Appendices C, E, and G |
| SCD | Universal algorithm | Smallest-consistent-candidate selection, adapted from the critical-language idea of Kleinberg & Mullainathan (2024). | Appendix D |
| MQ | Universal algorithm | Membership-query comparison of possible input-output pairs, adapted from Angluin (1988). | Appendices C and G |
| **Arguments developed for the present map setting** | | | |
| FA | Positive bridge | Finite active access reveals every output in a finite output set and certifies when that set is empty. | Appendix C and the positive finite-output proofs |
| NI | Impossibility | Missing negative information: enumerating valid candidate outputs does not show that an unseen output is invalid. | Appendices C and G |
| FI | Impossibility | Fresh-input ambiguity: after finitely many candidate queries, an untested output may still be valid at a newly requested input. | Appendix D |
| IO | Impossibility | Infinite-output ambiguity: a finite part of an infinite enumeration remains compatible with different unseen extensions. | Appendix G |
| CT | Universal algorithm | Candidate answer and target check: an active candidate proposes an answer, which is then checked through target membership queries. | Prediction and verification in Appendix G |
| **Direct special cases and reused lower bounds** | | | |
| SV | Universal algorithm | Direct exhaustive comparison for total single-valued maps. | Appendix F |
| Sub[X] | Reused impossibility | The possibly-infinite-output class contains all finite-output maps, so the finite-output lower bound X applies directly. | Appendix G |

For example, consider $(P, A)$ in the finite-output table. The learner passively receives valid pairs from the target but can actively query the candidates. The code SCD+FA says that prediction is possible by combining smallest-consistent-candidate selection with the information supplied by active access to finite output sets. Identification has the code LI because its impossibility follows from classical language identification. Verification also has the code LI because positive target observations cannot determine whether the target has any output at a previously unseen input.

The background colors classify the arguments. Light red marks arguments taken from or adapted from earlier work, light green marks arguments developed for the present map setting, and light yellow marks direct special cases or lower bounds reused from a smaller class of maps. A combined code such as SCD+FA

is colored green because the result also requires the new finite-output argument. A code of the form Sub[X] is colored yellow because it reuses the lower bound X from the finite-output subclass. The colors classify the source of the argument, not whether the result is positive or negative.

Table 2: Argument prototypes for all 81 cases. Id., Pred., and Ver. denote identification, prediction, and verification, respectively. P, A, and Q refer to passive, active, and membership query (testing), respectively. The meaning and role of each code are given in Table 1.

| Access $(T, C)$ | Single-valued | | | Finite-output | | | Possibly-infinite-output | | |
|---|---|---|---|---|---|---|---|---|---|
| | Id. | Pred. | Ver. | Id. | Pred. | Ver. | Id. | Pred. | Ver. |
| $(P, P)$ | SV | SV | SV | LI | FI | FI | Sub[LI] | Sub[FI] | Sub[FI] |
| $(P, A)$ | SV | SV | SV | LI | SCD+FA | LI | Sub[LI] | IO | Sub[LI] |
| $(P, Q)$ | SV | SV | SV | LI | FI | FI | Sub[LI] | Sub[FI] | Sub[FI] |
| $(A, P)$ | SV | SV | SV | NI | FI | FI | Sub[NI] | Sub[FI] | Sub[FI] |
| $(A, A)$ | SV | SV | SV | MQ+FA | SCD+FA | MQ+FA | IO | IO | IO |
| $(A, Q)$ | SV | SV | SV | MQ+FA | FI | FI | LI | Sub[FI] | Sub[FI] |
| $(Q, P)$ | SV | SV | SV | NI | FI | FI | Sub[NI] | Sub[FI] | Sub[FI] |
| $(Q, A)$ | SV | SV | SV | MQ+FA | SCD+FA | MQ+FA | NI | CT | CT |
| $(Q, Q)$ | SV | SV | SV | MQ | FI | FI | MQ | Sub[FI] | Sub[FI] |

# C Proof of the finite-output identification theorem

*Proof.* We break the proof into three parts. In the first part, we show that map identification for finite-output operators is universally possible when the ground-truth map and the candidate models are both observed actively or via testing. In the second and third parts, we show that map identification is not universally possible when the ground-truth map or the candidate models are observed passively.

**Part I: Map identification for finite-output operators is universally solvable with active or testing observations.** First, note that any algorithm can mimic a testing observation with active observations. That is, suppose the algorithm wants to know whether $g \in \mathcal{M}f$ for some operator $\mathcal{M}$, $f \in \mathcal{X}$, and $g \in \mathcal{Y}$. Then, the algorithm can keep actively observing $\mathcal{M}$ with input $f$ and eventually the whole set $\mathcal{M}f$ is observed; at which point it can check if $g \in \mathcal{M}f$.

Hence, to show map identification is universally solvable for finite-output operators with active or testing observations, it suffices to assume that the algorithm observes both the ground-truth map and candidate models via testing. Let $(f_1, g_1), (f_2, g_2), \ldots$ be the fixed enumeration selected by the adversary. The algorithm begins by selecting $\mathcal{M}_1$ as its current best guess at $\mathcal{M}_*$. If in the $t$th round of the game the algorithm's best guess is $\mathcal{M}_k$, then the algorithm queries if $g_t \in \mathcal{M}_* f_t$ and if $g_t \in \mathcal{M}_k f_t$. If the algorithm finds that $g_t \in \mathcal{M}_* f_t$ and $g_t \notin \mathcal{M}_k f_t$ or $g_t \notin \mathcal{M}_* f_t$ and $g_t \in \mathcal{M}_k f_t$ (i.e., $g_t \in \mathcal{M}_* f_t \triangle \mathcal{M}_k f_t$, where $\triangle$ is the symmetric difference between two sets), then the algorithm discards $\mathcal{M}_k$ and moves on to $\mathcal{M}_{k+1}$ as its current best guess. The round counter is set back to $t = 1$ and the game continues. (The round counter is set back to 1 so that the algorithm can test $\mathcal{M}_{k+1}$ with $f_1, f_2, \ldots, .$)

To show that the algorithm wins the game, assume that $\mathcal{M}_* = \mathcal{M}_{k_*}$ for some $k_*$. Then, for every $k < k_*$, since $\mathcal{M}_k \neq \mathcal{M}_*$, there must exist some $f_t$ and $g_t$ for which $g_t \in \mathcal{M}_* f_t \triangle \mathcal{M}_k f_t$. Hence, the algorithm discards $\mathcal{M}_k$ as its current best guess. That means the algorithm eventually selects $\mathcal{M}_{k_*}$ and never discards it since $\mathcal{M}_{k_*} = \mathcal{M}_*$, winning the map identification game.

**Part II: Map identification for finite-output operators is not possible with passive observations of the ground-truth.** In part II of the proof, we play the map identification game with an algorithm that passively observes the ground-truth map. We show that map identification is not universally solvable, regardless of how the candidate models are observed. Since the algorithm can mimic a passive observation or perform a test with active observations, it suffices to assume that the algorithm observes the candidate models actively.

Suppose that there is an algorithm that universally solves the map identification. We now show that such an algorithm can universally solve the set identification problem, which contradicts Gold's 1967 result (Gold, 1967). Given any set identification problem, let $\mathcal{X} = \bigcup_{i=1}^{\infty} S_i$ and $\mathcal{Y} = \{1\}$, respectively. We define the ground-truth map and candidate models as indicators of the sets:

$$\mathcal{M}_* s = \begin{cases} \emptyset, & \text{if } s \notin S_*, \\ \{1\}, & \text{if } s \in S_*, \end{cases} \qquad \mathcal{M}_i s = \begin{cases} \emptyset, & \text{if } s \notin S_i, \\ \{1\}, & \text{if } s \in S_i. \end{cases} \tag{2}$$

A passive observation of $\mathcal{M}_*$ is equivalent to obtaining an element $s_k \in S_*$ and an active observation of $\mathcal{M}_i$ with an input $s$ is equivalent to asking if $s \in S_i$. Since the algorithm universally wins the map identification game, the algorithm correctly identifies $\mathcal{M}_*$ after a finite number of rounds. Moreover, $\mathcal{M}_*$ is an indicator for $S_*$, so the algorithm identifies which $S_j = S_*$ and universally solves the set identification problem, leading to a contradiction.

**Part III: Map identification for finite-output operators is not universally possible with passive observations of the candidate models.** In part III of the proof, we play the map identification game with an algorithm that passively observes the candidates. We show that map identification is not universally possible, regardless of how the ground-truth map is observed. Since the algorithm can mimic an active observation or perform a test with active observations, it suffices to assume that the algorithm actively observes the ground-truth map. We again suppose that a universal algorithm exists and seek a contradiction.

Assume, for the sake of contradiction, that there exists a universal algorithm for map identification when the ground-truth map is observed actively while candidate models are observed only passively. Let the input

space be $\mathcal{X} = \mathbb{Z}$ and the output space be $\mathcal{Y} = \{1\}$. For notational convenience, identify each candidate model $\mathcal{M}_i$ with a subset $S_i \subset \mathbb{Z}$ by setting

$$\mathcal{M}_i(n) = \begin{cases} \{1\}, & \text{if } n \in S_i, \\ \emptyset, & \text{otherwise.} \end{cases}$$

The adversary controls the passive observations of candidate models as follows. For any candidate $\mathcal{M}_i$, if no observation has yet been made, the algorithm receives the input-output pair $(0, \{1\})$. Otherwise, two types of passive observations are provided:

- A "top" observation reveals the input-output pair $(j + 1, \{1\})$, where $j$ is the largest input observed for $\mathcal{M}_i$ so far.
- A "bottom" observation reveals the input-output pair $(j - 1, \{1\})$, where $j$ is the smallest input observed.

We partition the execution into stages and set up an infinite sequence of map identification problems (all with $\mathcal{M}_*(n) = \{1\}$ for all $n \in \mathbb{Z}$), so that the first $k$ of them look identical to the algorithm until the end of the $k$th stage. In the $k$th stage:

- For candidate $\mathcal{M}_1$, the first passive observation is taken from the top, while all subsequent observations in that stage are taken from the bottom.
- For every candidate $\mathcal{M}_j$ with $j \geq 2$ and $j \neq k + 1$, all passive observations are from the bottom.
- For candidate $\mathcal{M}_{k+1}$, passive observations alternate between top and bottom.

Since the algorithm is assumed to be universally correct, it must eventually commit to a candidate in each stage. In particular, during stage $k$, it must eventually declare that $\mathcal{M}_* = \mathcal{M}_{k+1}$; otherwise, the adversary may simply set $\mathcal{M}_{k+1}$ equal to the ground truth, ensuring the algorithm never succeeds.

Let $i_k$ denote the largest input for which a passive observation has been received from $\mathcal{M}_{k+1}$ by the end of stage $k$. The adversary then redefines the candidate models by setting

$$\mathcal{M}_1(n) = \{1\}, \quad n \in \mathbb{Z}, \qquad \mathcal{M}_{k+1}(n) = \begin{cases} \{1\}, & n \leq i_k, \\ \emptyset, & n > i_k. \end{cases}$$

Under this construction, the ground truth map remains $\mathcal{M}_1$, while the algorithm's guess, $\mathcal{M}_{k+1}$, is a proper subset of $\mathbb{Z}$. Consequently, $\mathcal{M}_{k+1} \neq \mathcal{M}_*$. Since the adversary can repeat this procedure at every stage, an infinite sequence of map identification problems is generated in which the algorithm is incorrect in every stage. This contradiction shows that no universal algorithm can succeed when candidate models are observed only passively.

$\square$

# D Proof of the finite-output prediction theorem

*Proof.* We break the proof into two parts.

**Part I: Map prediction is universally possible with active observations of the candidate models.**
First, we note that an algorithm can mimic a passive observation of a map $\mathcal{M}$ by making finitely many active observations of it. To do so, given an enumeration of $\mathcal{X} = \{f_1, f_2, \ldots\}$, we can make active observations with inputs $f_1, f_2, \ldots$, until we see an element $g \in \mathcal{M}f_j$ for some $j \geq 1$. We then take $(f_j, g)$ as a passive observation. Hence, it suffices to assume that the algorithm collects passive observations of the ground-truth map and active observations of candidate models.

Let $(f_{j_1}, g_{j_1}), (f_{j_2}, g_{j_2}), \ldots$ be the enumerated passive observations of the ground-truth map $\mathcal{M}_*$. In the $t$th round of the map prediction game, let $f_{\ell_t}$ be the "adversary's selected input." We define $\mathcal{C}_t$ to be the set of feasible candidate models, where $\mathcal{M}_j \in \mathcal{C}_t$ if the following three conditions hold:

1. (finiteness) $1 \leq j \leq t$,

2. (consistent) $g_{j_i} \in \mathcal{M}_j f_{j_i}$ for every $1 \leq i \leq t$ and $\mathcal{M}_j f_{\ell_t} \neq \emptyset$,

3. (minimality) for every $\mathcal{M}_{i'}$ such that $1 \leq i' \leq j$, $g_{j_i} \in \mathcal{M}_{i'} f_{j_i}$ for every $1 \leq i \leq t$, and $\mathcal{M}_{i'} f_{\ell_t} \neq \emptyset$, we have that $\mathcal{M}_j f_{\ell_t} \subset \mathcal{M}_{i'} f_{\ell_t}$.

Since we assume that $\mathcal{M}_j f$ is a set with finite cardinality for all $j \geq 1$ and $f \in \mathcal{X}$, the algorithm can compute $\mathcal{C}_t$ by making finitely many active observations of the candidate models. If $\mathcal{C}_t$ is empty, then we return an arbitrary element of $\mathcal{Y}$ as our guess for an object in $\mathcal{M}_* f_{\ell_t}$. Otherwise, let $\mathcal{M}_{j_t}$ be the largest-indexed operator in $\mathcal{C}_t$. By the consistency property, we know that $\mathcal{M}_{j_t} f_{\ell_t}$ is non-empty. We design the algorithm to make an active observation of $\mathcal{M}_{j_t}$ with the input $f_{\ell_t}$ and return an arbitrary output from $\mathcal{M}_{j_t} f_{\ell_t}$ as our current best guess for $\mathcal{M}_* f_{\ell_t}$.

To prove the algorithm always selects a $g \in \mathcal{M}_* f_{\ell_t}$ in the "prediction" step after a finite number of rounds, we suppose that $\mathcal{M}_* = \mathcal{M}_{j_*}$ for some $j_* \geq 1$. First, we show that after some $t_0 \geq 1$, we have $\mathcal{M}_{j_*} \in \mathcal{C}_t$ for every $t \geq t_0$. To see this, for every $j < j_*$, if $\mathcal{M}_{j_*}$ is not a restriction of $\mathcal{M}_j$ at every $f \in \mathcal{X}$, then there exists some $f \in \mathcal{X}$ and $g \in \mathcal{Y}$ such that $g \in \mathcal{M}_{j_*} f$ but $g \notin \mathcal{M}_j f$. Eventually, the algorithm passively observes $(f, g)$ from the ground-truth. Hence, each $\mathcal{M}_j$ that appears before $\mathcal{M}_{j_*}$ either extends $\mathcal{M}_{j_*}$, i.e., $\mathcal{M}_{j_*} f \subset \mathcal{M}_j f$ for every $f \in \mathcal{X}$, or eventually violate the consistency property. That makes $\mathcal{M}_{j_*}$ satisfy the minimality assumption after finitely many rounds. Clearly, $\mathcal{M}_{j_*}$ always satisfies the consistency property and the finiteness property by taking $t_0 \geq j_*$. This proves that $\mathcal{M}_{j_*} \in \mathcal{C}_t$ for every $t \geq t_0$. Hence, at the $t$th iteration for any $t \geq t_0$, if we let $\mathcal{M}_k$ be the largest-indexed operator in $\mathcal{C}_t$, then by the minimality assumption, we must have that $\mathcal{M}_k f_{\ell_t} \subset \mathcal{M}_{j_*} f_{\ell_t} = \mathcal{M}_* f_{\ell_t}$. This shows that the algorithm makes the map prediction with $\mathcal{M}_*$ possible.

**Part II: Map prediction is not universally possible without actively observing the candidate models.** As explained earlier, an algorithm can mimic a passive observation by testing and mimic a passive observation or perform a test by making active observations. Hence, all we have to show is an algorithm that collects active observations of the ground-truth map and performs tests of the candidate models cannot universally learn an operator. Seeking a contradiction, assume an algorithm exists. We set $\mathcal{X} = \mathbb{N}_{\geq 0}$ and $\mathcal{Y} = \mathbb{N}$. We will consider an execution of the algorithm on an infinite sequence of map prediction problems, so that the first $k$ of them look identical to the algorithm until the end of the $k$th iteration. We then construct an additional map prediction problem that cannot be solved by the algorithm provided that the algorithm correctly solve the sequence of map prediction problems we defined. Throughout our construction, we assume that for any operator $\mathcal{M}$, we have that $\mathcal{M}0 = \{1\}$. Hence, in the following proof, we only need to specify an operator on $\mathbb{N}$.

- **Iteration 1:** As the base case, in the first iteration, when the algorithm actively observes the ground-truth map with input $k$ for some $k \geq 1$, they always get the empty set $\emptyset$ so that $\mathcal{M}_* k = \emptyset$. At some

point, the algorithm must stop and take the adversary's question.[1] Let $k_1$ be the largest input that has been actively observed by the algorithm. If the algorithm does not make any observation of the ground truth, then we set $k_1 = 0$. The adversary asks the algorithm to compute an output from $\mathcal{M}_* f_{f_1}$, where $f_1 = k_1 + 1$. Set $s_1 = 2$. Consider the following problem, where $\mathcal{M}_* = \mathcal{M}_{s_1}$ and $g \in \mathcal{M}_i f$ if and only if $i = s_1$, $f = f_1$, and $g = 1$. At some point, the algorithm must stop testing and guess $g_1 \in \mathcal{M}_* f_1$ for some $g_1 \geq 1$. Let $h_1$ be an element of $\mathcal{Y}$ that is not equal to $g_1$ and has not been tested by the algorithm so far.

- **Iteration k:** Assume that for every $1 \leq t \leq k - 1$, $f_t$ is the input that the adversary picked in the $t$th iteration and the algorithm guessed that $g_t \in \mathcal{M}_* f_t$. Suppose that $h_t \neq g_t$ for every $1 \leq t \leq k - 1$ and that $h_t$ was not tested by the algorithm until the end of the $t$th iteration. In the $k$th iteration, when the algorithm actively observes $\mathcal{M}_*$ with input $f$ for some $f \geq 1$: if $f = \ell_t$ for some $t \leq k - 1$, we return $h_t$ as the only active observation; otherwise, the algorithm gets that $\mathcal{M}_* f = \emptyset$. At some point, the algorithm must stop observing the ground-truth map and let the adversary ask a question. We set $f_k > f_{k-1}$ to be an input that has never been observed or tested by the algorithm so far and let the adversary ask for an element of $\mathcal{M}_* f_k$. Set $s_k > s_{k-1}$ to be an index so that $\mathcal{M}_{s_k}$ has never been tested by the algorithm. Consider the following map prediction problem, where

$$\mathcal{M}_1 f = \begin{cases} \{h_t\}, & f = f_t \text{ for some } t \leq k - 1, \\ \emptyset, & \text{otherwise,} \end{cases}$$

for any $1 \leq t \leq k$, we have

$$\mathcal{M}_{s_t} f = \begin{cases} \{h_\tau\}, & f = f_\tau \text{ for some } \tau \leq t - 1, \\ \{1\}, & f = f_t, \\ \emptyset, & \text{otherwise,} \end{cases}$$

and $\mathcal{M}_i f = \emptyset$ for any other $\mathcal{M}_i$ and $f \geq 1$. In this game, we set $\mathcal{M}_* = \mathcal{M}_{s_t}$. Note that up to the end of $(k - 1)$st iteration:

- For every candidate model $\mathcal{M}_i$ and every input $f$ such that $\mathcal{M}_i f$ has been tested by the algorithm, the output $\mathcal{M}_i f$ does not change from the $(k - 1)$st problem to the $k$th problem.
- For every input $f$ such that $\mathcal{M}_* f$ has been actively observed by the algorithm, the output $\mathcal{M}_* f$ does not change from the $(k - 1)$st problem to the $k$th problem.

Hence, when the algorithm is executed on the $k$th problem, it follows exactly the first $(k - 1)$ iterations that we defined and, after making finitely many tests in the $k$th iteration, will guess that $g_k \in \mathcal{M}_* f_k$ for some $g_k \in \mathcal{Y}$. Let $h_k$ be an element of $\mathcal{Y}$ that is not equal to $g_k$ and has not been tested by the algorithm so far.

- The additional problem: Consider a problem such that

$$\mathcal{M}_1 f = \begin{cases} \{h_k\}, & f = f_k \text{ for some } k \geq 1, \\ \emptyset, & \text{otherwise,} \end{cases}$$

for some $k \geq 1$, we have

$$\mathcal{M}_{s_k} f = \begin{cases} \{h_t\}, & f = f_t \text{ for some } t \leq k - 1, \\ \{1\}, & f = f_k, \\ \emptyset, & \text{otherwise,} \end{cases}$$

and $\mathcal{M}_i f = \emptyset$ for any other $\mathcal{M}_i$ and $f \geq 1$. We set $\mathcal{M}_* = \mathcal{M}_1$. When the algorithm is executed, it is easy to check that it follows exactly the sequence of iterations we defined above, guessing that $g_k \in \mathcal{M}_* f_k$ for every $k \geq 1$, which is always wrong. Hence, we see that the algorithm fails on the additional problem and reach a contradiction.

$\square$

---

[1]Otherwise, the algorithm never stops on a problem where $\mathcal{M}_* = \mathcal{M}_1$ is the operator so that $\mathcal{M}_* 0 = \{1\}$ and $\mathcal{M}_* j = \emptyset$ for all $j \geq 1$.

# E   Proof of the finite-output verification theorem

*Proof.* If an algorithm can identify a map, then it is obvious that it can also learn the operator as long as it makes active observations of the candidate models. Hence, from Theorem 1, the "if" direction is proved. Moreover, since it is obvious that a map prediction problem is easier than a map verification problem, from Theorem 2, we know that map verification is impossible when the algorithm makes passive observations or performs tests of the candidates. Then, the only thing that remains to be done is to show that map verification problem is impossible when the algorithm makes passive observations of the ground-truth and active observations of the candidates. To this end, we will show that if such an algorithm exists, then we can use it to solve the set identification problem (see the proof of Theorem 1), which leads to a contradiction. Let $S_1, S_2, \ldots$ be a sequence of sets such that each $S_j \subset \mathcal{X}$ is a subset of a countable set $\mathcal{X}$. Let $s_1, s_2, \ldots$ be an enumeration of $\mathcal{X}$ and let $s_{k_1}, s_{k_2}, \ldots$ be the enumeration of the ground-truth $S_* = S_{j_*}$ that we will passively observe. We set $\mathcal{Y} = \{1\}$ define a sequence of operators on $\mathcal{X}$ by

$$\mathcal{M}_i s = \begin{cases} \{1\}, & s \in S_i, \\ \emptyset, & s \notin S_i. \end{cases}$$

Then, $\{(s_{k_i}, 1)\}_{i=1}^{\infty}$ is an enumeration of solutions of $\mathcal{M}_* := \mathcal{M}_{j_*}$. In our map verification algorithm, we let the adversary pick

$$f^{(1)}, f^{(2)}, \ldots = \underbrace{s_1}_{B_1}, \underbrace{s_1, s_2}_{B_2}, \underbrace{s_1, s_2, s_3}_{B_3}, \ldots, \underbrace{s_1, \ldots, s_n}_{B_n}, \ldots$$

as the queried inputs. When the algorithm makes an active observation of $\mathcal{M}_i$ with input $s$, we use the set identification algorithm to check if $s \in S_i$. The algorithm will see that $\mathcal{M}_i s = \{1\}$ if $s \in S_i$ and $\mathcal{M}_i s = \emptyset$ otherwise. After the algorithm answers the adversary's questions in a full $B_n$, the set identification algorithm guesses $S_*$ to be $S_i$ with the smallest $i$ such that $1 \leq i \leq n$ and $S_i \cap \{s_j\}_{j=1}^n$ are exactly the inputs in $B_n$ of which the algorithm guesses an output of $1 \in \mathcal{Y}$. If no such set exists, then return an arbitrary one. By our assumption of the map verification algorithm, there exists some $n_0 \geq 1$ such that we always make the right guess for the entire $B_n$ for any $n \geq n_0$. For any $j < j_*$, since $S_j \neq S_{j_*}$, there exists an element $s_{d_j} \in S_j \triangle S_{j'}$. Set $n_1 = \max\{d_1, \ldots, d_{j_*-1}, n_0\}$. Then, for any $n \geq n_1$, we consider what happens after the full $B_n$ is answered by the algorithm. For every $j \leq j_*$, either $S_j$ contains $s_{d_j}$ but the algorithm guesses that $\mathcal{M}_* s_{d_j} = \emptyset$ or $S_j$ does not contain $s_{d_j}$ but the algorithm guesses that $1 \in \mathcal{M}_* s_{d_j}$. On the other hand, $S_{j_*}$ always contains exactly those of $s_1, \ldots, s_n$ for which the algorithm guesses 1. That means our set identification algorithm will always return the correct ground-truth $S_{j_*}$ after $B_{n_1}$ is encountered, which is a contradiction to the fact that there exists no set identification algorithm. $\qquad\square$

# F   Proof of the single-valued classification

*Proof.* We prove the claim in two parts. In Part I we show that map identification is universally solvable for single-valued maps when both the ground-truth map and the candidate models are observed passively. In Part II, we explain why the same conclusion holds regardless of whether the observations are passive, active, or via testing and why map prediction and map verification also follow immediately.

**Part I: Map identification is universally solvable with passive observations.** Since every operator $\mathcal{M}$ is single-valued, for every $f \in \mathcal{X}$ the output $\mathcal{M}f$ is a singleton set. The algorithm proceeds as follows. It begins by setting its current candidate guess to $\mathcal{M}_1$. We then partition the execution into stages, where the $k$th stage is dedicated to testing whether $\mathcal{M}_k$ is equal to $\mathcal{M}_*$. During each round of the game in the $k$th stage, the algorithm passively observes a new input $f$ and records the corresponding two outputs: (i) $g \in \mathcal{M}_*f$ and (ii) $\hat{g} \in \mathcal{M}_kf$.

Let $\mathcal{D}_*$ and $\mathcal{D}_k$ denote the sets of observed input-output pairs for $\mathcal{M}_*$ and $\mathcal{M}_k$, respectively. If for some input $f$ the outputs differ, i.e., $g \neq \hat{g}$, then the algorithm concludes that $\mathcal{M}_k \neq \mathcal{M}_*$ and immediately discards $\mathcal{M}_k$, moving on to the next candidate $\mathcal{M}_{k+1}$, terminating stage $k$ and entering stage $k + 1$.

Now, suppose that $\mathcal{M}_* = \mathcal{M}_{k_*}$ for some $k_* \geq 1$. Then, for every candidate $\mathcal{M}_k$ with $k < k_*$ the single-valuedness of the maps guarantee that there exists at least one input $f \in \mathcal{X}$ for which

$$\mathcal{M}_*f \neq \mathcal{M}_kf.$$

Since a passive observation requires the inputs to be enumerated (see Definition 2), it will take at most a finite number of rounds before the algorithm will observe a discrepancy and discard $\mathcal{M}_k$. When stage $k_*$ is reached, the algorithm will correctly identify the map in each subsequent round and no discrepancy will ever be observed. Therefore, the algorithm eventually selects the correct candidate model and remains with it thereafter.

**Part II: Extension to all types of observations and all ML goals.** The above algorithm assumes passive observations for both the ground-truth map and candidate models. However, if the observations are made actively or via testing, the algorithm can easily mimic passive observations. Hence, map identification is universally possible for any types of observations. Furthermore, because map prediction and map verification are essentially equivalent to map identification for single-valued maps, all ML goals are universally solvable under any types of observations. □

# G   Proof of the possibly-infinite-output classification

*Proof.* We break our proof into three parts, corresponding to the three statements of the theorem, respectively.

**Part I: map identification.** To show the "if" direction, we can apply exactly the same algorithm proposed in Theorem 1 when the algorithm tests the ground-truth and candidate models. To prove the "only if" direction, we first note that since the finite-output map identification problem is easier than the infinite-output one, Theorem 1 immediately says that infinite-output map identification is impossible when the algorithm makes passive observation of the ground-truth or the candidate models. Hence, it remains to show that map identification is impossible when the ground-truth is actively observed and the candidate models are actively observed or tested, or when the ground-truth is tested and the candidate models are actively observed. Since for infinite-output operators, neither an active observer nor a test performer is strictly stronger than the other, we have to prove the three cases separately.

**I(i):  map identification is impossible when the algorithm makes active observations of the ground-truth and the candidate models.** We defer this case to part II, because when the algorithm can actively observe the candidate models, the map prediction problem is clearly easier than the map identification problem; proving the infeasibility of winning the former problem will also do the same for the latter one.

**I(ii):  map identification is impossible when the algorithm makes active observations of the ground-truth and performs tests of the candidate models.** If such an algorithm exists, we will use it to solve the set identification problem defined in the proof of theorem 1. Let $S_1, S_2, \ldots$ be a sequence of distinct candidate sets and let $S_* = S_{j_*}$ be a ground-truth set that is equal to one of the candidates. Without loss of generality, we assume that $S_j$ is countably infinite for every $j \geq 1$. Let $s_1, s_2, \ldots$ be an enumeration of $S_*$. We define $\mathcal{X} = \{f_1, f_2, \ldots\}$ and $\mathcal{Y} = \bigcup_{j=1}^\infty S_j$. Imagine that the candidate operators $\mathcal{M}_j$ are defined by

$$\mathcal{M}_j f_1 = S_j, \quad \mathcal{M}_j f_i = \emptyset, \qquad j \in \mathbb{N}, \quad i \geq 2,$$

and the ground-truth operator is $\mathcal{M}_{j_*}$. We apply the map identification algorithm: whenever the algorithm makes an active observation of the ground-truth $\mathcal{M}_*$ with some $f_i$, if $i \geq 2$, we return the emptyset; otherwise, if $i = 1$, we query an element $s$ from $S_*$ and answer that $s \in \mathcal{M}_* f_1$. When the algorithm tests if $s \in \mathcal{M}_j f_i$, we answer yes if and only if $i = 1$ and $s \in S_j$, which is determined from the set identification algorithm. Let the set identification algorithm guess $S_j$ for the $j \in \mathbb{N}$ such that $\mathcal{M}_j$ is guessed by the algorithm in the map identification game. Clearly, the algorithm guesses correctly if and only if the set identification algorithm guesses correctly. We reach a contradiction.

**I(iii):  map identification is impossible when the algorithm performs tests of the ground-truth and makes active observations of the candidate models.**

In this section, we first consider the following question. Let $S_1, S_2, \ldots$ be subsets of $\mathbb{N}$, exactly one of which equals $\mathbb{N}$. An adversary enumerates every $S_j$, and a query to $S_j$ returns the next unseen element in its enumeration. Is there an algorithm that identifies which $S_j$ equals $\mathbb{N}$? As before, the game is played in rounds: the algorithm makes finitely many queries and then returns a guess, and identification means that the guess is always correct after some finite round.

It is easy to see that one can reduce the set identification game to a map identification game with test/ground-truth and active/candidates. Namely, we consider a problem where $\mathcal{X} = \{\star\}$ is a singleton and $\mathcal{M}_j \star = S_j$, enumerated by the adversary in the same way as in the set identification game. Let $\mathcal{M}_* \star = \mathbb{N}$. (We do not even need to let the algorithm test the ground-truth. We can directly tell the algorithm that $\mathcal{M}_* \star = \mathbb{N}$.) When the map identification algorithm makes an active observation of a candidate $\mathcal{M}_j \star$, it is as if a set identification algorithm asks for an element of $S_j$. Hence, if the map identification algorithm can identify which $\mathcal{M}_j$ is equivalent to $\mathcal{M}_*$, then we can use this algorithm to identify which $S_j$ is equivalent to $\mathbb{N}$.

Now, we will show that the set identification problem is infeasible, which shows that the map identification problem (whose solution, if exists, would lead to a set identification algorithm) is infeasible either. It works as follows:

1. First, we hold out an element, say $1 \in \mathbb{N}$. Let $S_1 = \{2, 3, 4, \ldots\}$ and $S_2 = \{1, 2, 3, \ldots\}$. When the algorithm is executed on this problem, it needs to say that $S_2 = \mathbb{N}$ at some point. At this moment, assume that $\{2, 3, \ldots, j_1\}$ are seen from $S_1$ and $\{1, 2, \ldots, k_1\}$ are seen from $S_2$. (Remark: without loss of generality, we can assume that the algorithm only queries $S_1$ and $S_2$ in this execution. If it queries another set, say $S_i$, then we can simply assume that $S_i = \emptyset$ and ignore it thereafter.)

2. Now, the idea is that we want to trick the algorithm by making $S_2$ the wrong answer. This is simple. In the next problem, we can simply set $S_2 = \{1, 2, \ldots, k_1\}$. We define $S_3 = \{1, 2, 3, \ldots\}$. (If $S_3$ is queried in the previous execution, then we use $S_4$ instead, and so on. See the remark above.) For $S_1$, we change its definition to $S_1 = \{2, 3, \ldots, j_1, 1, j_1 + 2, j_2 + 3, \ldots\} = \mathbb{N} \setminus \{j_1 + 1\}$. Intuitively, we add back 1 but hold out $j_1 + 1$. We add back 1 because in our final problem (the diagonalization), we want $S_1 = \mathbb{N}$ to be the correct answer; we hold out $j_1 + 1$ because in the current problem, we want $S_3 = \mathbb{N}$ to be the correct answer. When the algorithm is executed, it is clear that it follows the first stage up to the point that both $j_1 \in S_1$ and $k_1 \in S_2$ are seen. After that, if the algorithm queries $S_2$ again, then we will loop back to 1 in this setting and output $k_1 + 1$ in the previous setting, but this no longer matters. The thing is that at some point, the algorithm will guess that $S_3 = \mathbb{N}$. At this moment, assume that $\{2, 3, \ldots, j_1, 1, j_1 + 2, j_1 + 3, \ldots, j_2\}$ are seen from $S_1$ and $\{1, 2, \ldots, k_2\}$ are seen from $S_3$.

3. At this point, you can expect what happens in the next problem. We set $S_2 = \{1, 2, \ldots, k_1\}$ and $S_3 = \{1, 2, \ldots, k_2\}$. Let $S_4 = \mathbb{N}$ be the new ground-truth. For $S_1$, we add back $j_1 + 1$ but hold out $j_2 + 1$. That is, its enumeration would be

$$S_1 = \{\underbrace{2, 3, \ldots, j_1,}_{\text{first stage}} \underbrace{1, j_1 + 2, j_1 + 3, \ldots, j_2,}_{\text{second stage}} j_1 + 1, j_2 + 2, j_2 + 3, \ldots\}.$$

The rest follows the same pattern. Intuitively, why this would lead to a contradiction is that whenever you hold out an element of $\mathbb{N}$ from $S_1$, you will always add it back in the next problem; hence, in the infinite limit, no element is left out and we thus construct an enumeration of $S_1 = \mathbb{N}$.

Now, it all remains to define a final problem, for which $S_1 = \mathbb{N}$, but the algorithm guesses incorrectly infinitely often. To do this, let $S_j = \{1, 2, \ldots, k_{j-1}\}$ $(j \geq 2)$ be whatever the finite set defined in the $j$th stage. Consider the following enumeration of $S_1$: first, let $B_i = \{j_i, \ldots, j_{i+1} - 1\}$ be the block defined by

$$\underbrace{1(= j_0), 2, \ldots, j_1 - 1,}_{B_0} \underbrace{j_1, j_1 + 1, \ldots, j_2 - 1,}_{B_1} \underbrace{j_2, j_2 + 1, \ldots, j_3 - 1,}_{B_2} j_3 \ldots.$$

Consider the enumeration of $B_i$: $B_i' = \{j_i + 1, \ldots, j_{i+1} - 1, j_i\}$ (i.e., rotating the first element to the end). Then, $B_0', B_1', B_2', \ldots$ is apparently an enumeration of $S_1 = \mathbb{N}$. However, when executing the algorithm on this enumeration, we follow exactly the sequence of stages defined above, guessing that $S_2, S_3, \ldots$ are the ground-truth. That is, the algorithm makes incorrect guesses infinitely often.

Assume such an algorithm exists. Let $\mathcal{X} = \{f\}$ be a singleton and let $\mathcal{Y} = \mathbb{N}$. We partition the execution into stages, each of which contains multiple iterations of the game, and set up an infinite sequence of map identification problems, so that the first $k$ of them look identical to the algorithm until the end of the $k$th stage. For each of these map identification problems, we assume that the ground-truth operator is $\mathcal{M}_*$ such that $\mathcal{M}_* f = \mathbb{N}$. Then, we define an additional map identification problem and show that the algorithm must fail to solve it provided that it solves the sequence of problems we defined.

- **Stage 1:** Set $s_1 = 2$ and $j_1 = 0$. We define the candidate models to be defined by

$$\mathcal{M}_i f = \begin{cases} \mathbb{N} \setminus \{j_i + 1\}, & i = 1, \\ \mathbb{N}, & i = s_1, \\ \emptyset, & \text{otherwise.} \end{cases}$$

Let $j_1 + 2, j_1 + 3, \ldots$ be the adversary's enumeration of $\mathcal{M}_1 f$ and let $1, 2, \ldots$ be the adversary's enumeration of $\mathcal{M}_{s_1} f$. Since the algorithm is correct, at some iteration, it must guess that $\mathcal{M}_* = \mathcal{M}_{s_1}$. After this happens, we stop the first stage and enter the second stage.

- **Stage k:** Assume that the algorithm guesses $\mathcal{M}_* = \mathcal{M}_{s_t}$ in the $t$th stage for every $1 \le t \le k-1$. Let $s_k > s_{k-1}$ be an index for which $\mathcal{M}_{s_k}$ has never been actively observed up to the end of stage $k-1$. Let $j_k \in \mathcal{Y}$ be the largest output that has been actively observed from $\mathcal{M}_1 f$ up to the end of stage $k-1$. If no active observation of $\mathcal{M}_1 f$ has been made yet, then we set $j_k = 0$. We define the $k$th map identification problem by choosing the candidate models as follows:

$$
\mathcal{M}_i f = \begin{cases} \mathbb{N} \setminus \{j_k\}, & i = 1, \\ \emptyset, & i \ge 1, i \notin \{s_1, s_2, \ldots\}. \end{cases}
$$

  For $i = s_t$, where $1 \le t \le k-1$, we define $\mathcal{M}_i f$ to be the finite set of all elements actively observed at the end of the $(k-1)$st stage. The adversary enumerates $\mathcal{M}_1 f$ by following the order that the set $\{1, \ldots, j_k\} \setminus \{j_{k-1}\}$ is enumerated in the first $(k-1)$ rounds, and then $j_{k-1}$, and then $j_k + 2, j_k + 3, \ldots$. Since the algorithm is correct, at some iteration, it must guess that $\mathcal{M}_* = \mathcal{M}_{s_k}$. After this happens, we stop the $k$th stage and enter the next ones.
- **The additional problem:** We define our additional problem by letting $\mathcal{M}_1 f = \mathbb{N}$ and $\mathcal{M}_i f = \emptyset$ for all $i \notin \{s_1, s_2, \ldots\}$. Given some $k \ge 1$, we note that the definition of $\mathcal{M}_{s_k} f$ is never changed after the $(k+1)$st stage. We take that as our definition of $\mathcal{M}_{s_k} f$ in our additional problem and the order that elements in $\mathcal{M}_{s_k} f$ are enumerated in the first $k$ stages as the adversary's enumeration of it. We enumerate $\mathcal{M}_1 f$ by following how $\mathbb{N}$ is enumerated in the sequence of stages, i.e.,

$$
\mathbb{N} = \{j_1 + 2, j_1 + 3, \ldots, j_2, j_1, j_2 + 2, j_2 + 3, \ldots, j_3, \ldots, j_k + 2, j_k + 3, \ldots, j_{k+1}, j_k, j_{k+1} + 2, \ldots\}.
$$

  When the algorithm is executed on this additional problem, it follows the sequence of stages we defined, returning $\mathcal{M}_{s_j}$ as its guess for every $j \ge 1$, which is false. Hence, the algorithm is not correct on the additional problem and we reached a contradiction.

**Part II: map prediction.** Similar to part I, we only need to consider the cases when the algorithm makes active observations of the candidate models since other cases follow from Theorem 2.

**II(i): map prediction is possible when the algorithm performs tests of the ground-truth and makes active observations of the candidate models.** We defer this case to part III, because a map verification problem is clearly harder than a map prediction problem. Hence, proving the feasibility of map verification problems in this setting will automatically prove the feasibility of map prediction problems.

**II(ii): map prediction is impossible when the algorithm makes passive or active observations of the ground-truth and makes active observations of the candidate models.** Since every finite-output problem is also regarded as an infinite-output problem in our setting, by Theorem 2, we only need to prove the impossibility when we actively observe the candidates and passively or actively observe the ground-truth. Moreover, since we can still use finitely many active observations to obtain a passive observation, even for infinite-output operators, it suffices to show that map prediction is impossible when the algorithm makes active observations of the ground-truth and candidate models. Assume such an algorithm exists. We let $\mathcal{X} = \mathcal{Y} = \mathbb{N}$. We partition the execution into stages, each of which contains multiple iterations of the game, and set up an infinite sequence of map prediction problems, so that the first $k$ of them look identical to the algorithm until the end of the $k$th stage. Each map prediction problem we define has a different ground-truth operator $\mathcal{M}_*$ and candidate models $\mathcal{M}_1, \mathcal{M}_2, \ldots$. Then, we use this sequence of map prediction problems to construct another problem so that when the algorithm is executed on this additional problem, it exactly follows the sequence of stages that we defined. We will show that as long as the algorithm correctly solves the sequence of problems, it must fail on the additional map prediction problem we construct, which leads to a contradiction. We now define the sequence of stages and map prediction problems inductively.

The high-level idea in this construction is the same as the one above: eventually, we want $\mathcal{M}_1$ to be the ground-truth; however, in every intermediate stage, we assume that a different candidate is the ground-truth. After the algorithm commits to that, in the following stages, we trick the algorithm and no longer make it the ground-truth.

In the first stage, consider $\mathcal{M}_1 i = $ all evens, $\mathcal{M}_2 i = $ all odds for all $i$, and $\mathcal{M}_* = \mathcal{M}_2$. The very important trick is that when the adversary picks an input $i$, $\mathcal{M}_* i$ has never been queried before. (Note that the order

in each iteration is query ground-truth → raise question → query candidates → answer.) Hence, at the point the algorithm eventually needs to guess that $i_1 \in \mathcal{M}_* j_1$ for some odd $i_1$, no element of the set $\mathcal{M}_* j_1$ has been revealed (this is very important). This opens room for cheating, because the next compatible stage can set $\mathcal{M}_* j_1$ to be the set of even numbers, which does not violate any information we have previously seen (only true because $\mathcal{M}_* j_1$ has not been queried up to this point). This makes the algorithm produce a guess that is inconsistent with $\mathcal{M}_1$ in the first stage.

The next stage follows basically the same pattern. To see how it works, we can simply do the same procedure (now with $\mathcal{M}_3$ instead of $\mathcal{M}_2$) on the untouched inputs. That is, suppose in the first stage, the first 10 elements have been queried (either from $\mathcal{M}_1$, $\mathcal{M}_2$, or $\mathcal{M}_*$) and $j_1 = 10$. Then, consider $\mathcal{M}_1 i = $ all evens, $\mathcal{M}_3 i = $ all odds for all $i \geq 11$, and $\mathcal{M}_* = \mathcal{M}_3$, and play exactly the same cheating game. The construction makes $\mathcal{M}_1$ and $\mathcal{M}_*$ agree on the first 10 inputs (while also maintaining the entire execution in the first stage; this is important) by setting $\mathcal{M}_1 j = \mathcal{M}_* j = \mathbb{N}$ for $j = 1, 2, \ldots, 9$ (after all, only finitely many elements were seen in stage 1, with all even numbers coming from $\mathcal{M}_1$ and all odd numbers from $\mathcal{M}_*$), and $\mathcal{M}_1 10 = \mathcal{M}_* 10 = $ all evens (this is okay because $\mathcal{M}_* 10$ is never queried in stage 1). After the algorithm needs to guess that $i_2 \in \mathcal{M}_* j_2$ for some odd $i_2$ and $j_2$ for which $\mathcal{M}_* j_2$ has never been queried, the next compatible stage sets $\mathcal{M}_* j_2$ to be the set of even numbers, which produces an inconsistency between $\mathcal{M}_1 j_2$ and the algorithm's guess.

After each stage, the construction makes $\mathcal{M}_1$ and $\mathcal{M}_*$ agree on all inputs that have been queried up to this point, and plays the cheating game on the untouched inputs. Since there are infinitely many stages, each input must belong to some stage, meaning that $\mathcal{M}_1$ and $\mathcal{M}_*$ must eventually agree on it. That means in the infinite limit, we will have $\mathcal{M}_* = \mathcal{M}_1$ (because they agree on every input; this is the diagonalization), but when the algorithm is executed, it is tricked infinitely often, leading to a contradiction.

- **Stage 1:** Set $s_1 = 2$. We define the first map prediction problem to have a ground-truth operator $\mathcal{M}_*$ so that $\mathcal{M}_* j$ is the set of odd natural numbers for every $j \in \mathcal{X}$. Define $\mathcal{M}_1$ so that $\mathcal{M}_* j$ is the set of all even natural numbers for every $j \in \mathcal{X}$, $\mathcal{M}_{s_1} = \mathcal{M}_*$, and $\mathcal{M}_i$ for $i \geq 3$ so that $\mathcal{M}_* j = \emptyset$ for every $j \in \mathcal{X}$. When the algorithm makes an active observation of the ground truth $\mathcal{M}_*$ with an input $j \in \mathcal{X}$, the outputs it receives form an enumeration of the odd numbers. Whenever it is the adversary's turn to ask a question, they pick an input $j \in \mathcal{X}$ that has not appeared in the algorithm's observations so far. We also assume the input picked by the adversary is larger than all inputs they have asked before. When the algorithm makes an active observation of $\mathcal{M}_1 j$ with any $j$, it receives an enumeration of even natural numbers. When the algorithm makes an active observation of $\mathcal{M}_{s_1} j$ with any $j$, it receives an enumeration of odd natural number. When the algorithm makes an active observation of any other operator with any input, it is informed that the output set is empty. Since the algorithm is correct, at some point, the algorithm must guess an odd number, say $i_1 \in \mathcal{M}_* j_1$ for some odd $i_1$. We terminate the first stage and enter the second stage.

- **Stage k:** Assume that the algorithm guesses that $i_t \in \mathcal{M}_* j_t$ in the $t$th stage for every $1 \leq t \leq k - 1$. Also let $s_t$ be the index such that $\mathcal{M}_* = \mathcal{M}_{s_t}$ in the $t$th map prediction game for every $1 \leq t \leq k - 1$. We define our $k$th map prediction problem as follows. Let $\mathcal{M}_*$ be defined by

$$\mathcal{M}_* j = \begin{cases} \mathbb{N}, & j \leq j_{k-1}, j \notin \{j_1, \ldots, j_{k-1}\}, \\ \text{the set of all even natural numbers}, & j \in \{j_1, \ldots, j_{k-1}\}, \\ \text{the set of all odd natural numbers}, & j > j_{k-1}. \end{cases}$$

Let $s_k$ be an index such that $s_k > s_{k-1}$ and $\mathcal{M}_{s_k}$ has never been observed by the algorithm in the first $(k-1)$ stages. Define $\mathcal{M}_1$ to be

$$\mathcal{M}_1 j = \begin{cases} \mathbb{N}, & j \leq j_{k-1}, j \notin \{j_1, \ldots, j_{k-1}\}, \\ \text{the set of all even natural numbers}, & \text{otherwise.} \end{cases}$$

Define $\mathcal{M}_{s_k} = \mathcal{M}_*$. For any other $i \neq 1, s_k$, we can simply define $\mathcal{M}_i$ to be the operator whose valid input-output pairs have all been actively observed by the algorithm in the first $(k-1)$ stages. Given this definition, when the algorithm executes, there is a way for the adversary to enumerate the active

observations so that the algorithm follows exactly the first $(k-1)$ stages. After that, whenever it is the adversary's turn to ask a question, they pick an input $j \in \mathcal{X}$ that has not appeared in the algorithm's observations so far. We also assume the input picked by the adversary is larger than all inputs they have asked before (including those asked in the first $k-1$ stages). In particular, that means for every $j \in \mathcal{X}$ that the adversary asks, we have $\mathcal{M}_* j$ is the set of all odd natural numbers. At some iteration, the algorithm must guess an odd number, say $i_k \in \mathcal{M}_* j_k$ for some odd $i_k > i_{k-1}$. We terminate the $k$th stage and enter the second stage.

- **The additional problem:** Now, we define an additional map prediction problem for which the algorithm must fail. We set $\mathcal{M}_* = \mathcal{M}_1$ to be

$$\mathcal{M}_* j = \mathcal{M}_1 j = \begin{cases} \mathbb{N}, & j \notin \{j_1, j_2, \ldots\}, \\ \text{the set of all even natural numbers,} & \text{otherwise.} \end{cases}$$

Note that we have $s_1 < s_2 < \cdots$. Hence, for any $i \geq 2$, we must have that $i < s_k$ for some $k \geq 1$. Note also that the definition of $\mathcal{M}_i$ has remained the same in the $t$th map prediction game for any $t \geq k$. We take that as the definition of $\mathcal{M}_i$ in our additional map prediction problem. It is easy to see that when the algorithm executes on this additional problem, it follows exactly the sequence of stages we defined, guessing $i_t \in \mathcal{M}_* j_t$ for $t = 1, 2, \ldots$, which is wrong since $i_t$ is odd. Hence, the algorithm is incorrect on the additional problem and we have reached a contradiction.

**Part III: map verification.** The "only if" direction follows immediately from the "only if" direction of part II. To show the "if" direction, we construct an algorithm for map verification when the algorithm performs tests of the ground-truth and makes active observations of the candidate models. We break our algorithm into stages, where each stage contains multiple iterations. We start with the first stage, and since the stages in this algorithm are mutually independent, we explain what the algorithm does in a single stage, which completes the description of the algorithm. At the beginning of the $k$th stage, we initialize the set $V_k = \emptyset$. In the $t$th iteration of the $k$th stage, when the adversary picks some $f \in \mathcal{Y}$, the algorithm makes an active observation of $\mathcal{M}_k$ with this $f$. If $\mathcal{M}_k f = \emptyset$, we add $f$ to $V_k$ and guess that $\mathcal{M}_* f = \emptyset$. Otherwise, let $g$ be an element of $\mathcal{M}_k f$. We guess that $g \in \mathcal{M}_* f$. In the $(t+1)$st iteration, we first test if $g_j \in \mathcal{M}_* f$ for all $j \leq t$ and $f \in V_k$. If we ever get a positive answer, then we terminate the $k$th stage and enter the $(k+1)$st stage. Also, if we guess that $g \in \mathcal{M}_* f$ in the $t$th iteration, then we also test if this is true. If not, we also terminate the $k$th stage and enter the $(k+1)$st stage. We claim that this algorithm is correct. To see this, in the $k$th stage, if we always answer the question correctly, then we are done. Otherwise, assume we answer $\mathcal{M}_* f$ incorrectly. There are two possibilities. The first is that we answer $g \in \mathcal{M}_* f$ but it is not. In that case, we will terminate the $k$th stage immediately when we test that $g \notin \mathcal{M}_* f$ in the next iteration. The second possibility is that we answer $\mathcal{M}_* f = \emptyset$ but it is actually not. Say, $g_j \in \mathcal{M}_* f$. Then, we must terminate the $k$th stage after the $(j+1)$st iteration. That is, as long as we make a wrong guess during the $k$th stage, we know that we will eventually move onto the $(k+1)$st stage. Assume $\mathcal{M}_* = \mathcal{M}_{k_*}$. If we enter the $k_*$th stage, we will never terminate it. Hence, the algorithm is correct. $\qquad\square$

