# OpenReview forum: "The Learnability of an Unknown System From Input-Output Data"
_TMLR — Under review for TMLR_

### Review · Reviewer_r92v · 2026-06-08

**Summary Of Contributions:**

The paper asks a genuinely interesting question: from the data we manage to collect, when is learning possible in principle and when isn't it? The stated hope is that the answer can serve as a guide for data-driven discovery in the sciences. Learning is formalized as recovering information about a map $\mathcal{M}: \mathcal{X} \to \mathcal{Y}$, with both spaces taken to be countably infinite. On top of this the authors build a vocabulary of learning tasks (map prediction, identification, verification) and observation types (passive, active, testing), and they study everything through a game in which an adversary feeds observations consistent with a ground-truth map and some candidate maps while an algorithm tries to finish the task in finitely many steps. The main results are theorems pinning down which tasks are solvable under which observations, for finite-output maps (e.g. Theorem 2), for the single-valued case (Section 6), and for the harder infinite-output case (Section 7).

I think the question is the real strength here. Asking about the fundamental limits of data-driven discovery is an ambitious and worthwhile thing to do, and it's worth saying that this kind of work is harder to pull off and harder to judge than a paper that proposes a method and runs it against baselines. The definitions of the observation and task types are clean and feel like a genuine contribution in their own right, and the writing is clear and the structure is sensible.

My reservations are mostly about narration rather than substance. The ideas are good but the paper underhits in selling them, and I'll unpack that in the boxes below: the main text reads a bit like a list of claims because the proofs and most of the intuition live in the appendix; a couple of the running examples drift away from the scientific framing; the countably-finite assumption sits awkwardly next to the "guide for science" pitch; and a few of the game definitions were hard to follow.

**Additional Comments:**

N/A

**Audience:**

Yes

**Audience Explanation:**

Yes, without much hesitation. TMLR has a real learning-theory and foundations readership, and asking when learning is even possible, and formalizing the kinds of observations and tasks that make it possible, is exactly the sort of question a good chunk of that audience cares about. Even setting the specific theorems aside, the task/observation taxonomy is something people working on the theory of data-driven science could pick up and build on.

**Claims And Evidence:**

No

**Claims Explanation:**

I want to be clear that this isn't me saying the math is wrong, the proofs are in the appendix and may well be fine. It's that the main text, on its own, doesn't yet make the results *clear* and *convincing*, and those are part of what this question is asking about.

The pattern throughout is a theorem followed immediately by "(see Appendix X)" with little intuition in between, so as a reader I can't really weigh the contribution without going to the appendix. Figure 2 gives a nice overview, but it summarizes claims rather than making me believe them. The problem is made worse by where the page budget goes: in a few spots, right when I'm expecting the implication or the intuition behind a theorem, I instead get an illustrative example, the passive/active image-tagging example at the end of Section 3 is the one that stuck with me. It's a fine example, it just shows up where I wanted the meaning of the result. The cumulative effect is that the paper loses some flow and some of its persuasiveness.

There are no experiments, which is completely fine for a paper like this, it doesn't claim any. But it does mean all the convincing has to come from the prose, and right now the prose isn't quite carrying it. A bit more walking-through of *why* an algorithm fails without active observations and then succeeds with them would make the theoretical evidence land much harder.

Two smaller things also got in the way of just following the claims. In the map-prediction game, the algorithm is asked to pick a "current best guess," but I couldn't find a precise statement of what "best guess" means, and I had a genuinely hard time playing the game out in my head as a result. And in the map-verification game the condition is written $g \in M_j$, which I'm fairly sure is a typo for $g \in M_j^*f$. Neither is fatal, but both made it harder to verify what's being claimed.

**Requested Changes:**

Things I'd consider critical for recommending acceptance:

The main one is bringing the intuition for each theorem into the body. A couple of sentences per result on what it means and why it holds would let a reader actually judge the contribution without living in the appendix, and it's the single change that would most move me on the evidence question.

I'd also flag two smaller things. The first is the "current best guess" in the map-prediction game, I suspect the authors find it obvious, and I did have something forming in my head, but unlike the task and observation concepts, which are laid out really carefully, this one isn't shown as explicitly, so a sentence making it precise would help. The second is just a likely typo: the map-verification condition reads $g \in M_j$, which I think should be $g \in M_j^*f$.

Things that would strengthen the paper but I don't consider blocking:

It would be really illuminating to see one of the separation results played out on a toy problem, for Theorem 2, say, a tiny algorithm that fails the map-prediction game without active observations and then solves it once active observations are allowed. No training or empirical setup needed; I just think watching the game resolve would make the whole framework click.

I'd also gently push on the examples. The infinite-output case is motivated with spam detection, but infinite-output problems are everywhere in science, weather forecasting and stochastic processes generally, particle unfolding, and so on, so it feels like a missed opportunity not to motivate it from there. I realize some of those live in continuous spaces and may bump against the countably-infinite assumption, which is exactly why I'd suggest, relatedly, a short paragraph acknowledging that assumption: many scientific domains are continuous ($\mathbb{R}^n$), and being upfront about what the framework does and doesn't cover would defuse the first objection most readers will have.

Finally, some light proofreading, e.g. "a inference" → "an inference", and trimming the occasional repetition (the Section 3 example again) to make room for the implications.

---

> ### Author Response · Authors · 2026-07-12
>
> We thank the reviewer for the careful review and insightful comments. We especially appreciate the distinction between concerns about the narration and concerns about the underlying mathematics. We have revised the paper so that readers can understand why the theorems hold without first reading every appendix proof.
>
> **Proof intuition in the main text:** Every theorem is now followed by a proof discussion/intuition describing the positive algorithm and the main obstruction in the negative cases. Theorem 1 explains exhaustive pair comparison and the missing-negative-data lower bounds. Theorem 2 explains both the smallest-consistent-candidate algorithm and the fresh-input ambiguity. Theorem 3 explains why verification adds an empty-output-set decision. Theorem 4 explains why unique outputs make positive data sufficient. Theorem 5 explains why infinite output sets require membership tests to obtain negative information. The original detailed proofs remain in Appendices C-G.
>
> **Organization and flow:** We have moved the extended applications out of the theorem sequence and into Appendix A. The compact overview figures remain in the main text, but the space beside the theorems is now used for proof ideas, interpretations, and comparisons between the three tasks. In particular, the repeated image-tagging discussion has been removed, and the implications of each theorem now follow the theorem directly.
>
> **Meaning of "current best guess":** The revised game definitions now state the exact output required in each round. Identification returns an index. Prediction returns an index $j$ and an element $g\in\mathcal{M}\_j f$. Verification returns a Boolean indicating whether $\mathcal{M}\_*f$ is empty and, for a nonempty answer, an index and witness. The prediction section also states explicitly that $(j,g)$ is only a provisional round output, not an optimizer under an unspecified loss or score. Correctness is determined solely by the eventual win condition.
>
> **Verification typo:** We thank the reviewer for catching this. The verification condition has been corrected to $g \in \mathcal{M}\_j f$, and the surrounding definition now states both the Boolean and witness requirements explicitly.
>
> **Worked separation for Theorem 2:** The discussion after Theorem 2 now plays out the requested finite interaction. At a fresh input $f$, suppose the learner tests finitely many proposed candidate outputs and receives only negative answers. The same finite transcript is compatible with an empty candidate output set and with a singleton output set whose only member has not yet been tested. Because every round must end after finitely many queries, testing cannot always resolve this ambiguity. Active candidate access resolves it by returning either a valid output or $\bot$. Appendix D turns this idea into the complete diagonal lower bound.
>
> **Scientific infinite-output example and countability:** We thank the reviewer for this suggestion. At the beginning of Section 7, we now briefly mention stochastic forecasting as a scientific setting that may produce countably many encoded future trajectories after quantization. We also clarify that continuous trajectories and probability distributions are not covered by the exact framework unless an additional countable representation is imposed. More broadly, Section 1.2 discusses this countability limitation, and Appendix A.7 illustrates how a fixed-resolution discretization can give an approximation interpretation in PDE learning.
>
> **Technical novelty of the work:** We have added a case-by-case comparison in Tables 1 and 2 of Appendix B to highlight the technical novelty of the proofs in the paper. Table 1 defines the argument prototypes, states whether each supports a universal algorithm or an impossibility result, gives its relationship to earlier work, and points to the complete proof. In particular, LI, SCD, and MQ identify arguments taken from or adapted from language identification, Kleinberg-Mullainathan generation, and membership-query learning. FA, NI, FI, IO, and CT identify arguments developed for the present input-output setting. SV and Sub[X] identify a direct single-valued argument and lower bounds reused from the finite-output subclass. Table 2 then assigns these codes to all 81 cases. Light red, green, and yellow backgrounds distinguish the three groups, and combined codes show when a conclusion uses more than one ingredient.
>
> **Minors:** We thank the reviewer for the careful read. We corrected "a inference," removed the repeated example, corrected and clarified the prediction and verification games, and proofread the surrounding notation and prose. The classification results and the original detailed proof constructions are unchanged.
>
> We hope this answers the reviewer's questions and concerns. We are happy to answer any follow-up question(s) that the reviewer may have.

---

> > ### Comment · Reviewer_r92v · 2026-07-16
> >
> > Many thanks to the authors for their careful consideration of the reviews.
> >
> > The paper reads better now with the proof sketches and the main concern from my review is addressed.
> >
> > Given the page limit, I understand that adding the proof ideas has shifted the practical implications away from the main text. I think that's fine, and I appreciate the application part in the appendix.
> >
> > The setting is still a strict one (countable spaces, etc.), but I find the contribution real. I would be happy to see the paper accepted, and thank the authors again.

---

### Review · Reviewer_r8xo · 2026-06-28

**Summary Of Contributions:**

The paper studies a very fundamental and a very general question: given input-output observations of an unknown system, when is it possible in principle to learn it? The problem is set up as an adversarial game between a learner, who holds a countable list of candidate maps (one of which is the true map M*), and an adversary, who fixes the order in which observations are revealed. The learner has one of three goals: 1. identify the map, 2. predict an output, or 3. verify an input. There are three ways to observe each of the inputs or outputs (passive, active, testing), leading to 27 settings, which are studied for three classes of map (single-valued, finite-output, infinite-output), giving a total of 81 cases, and for each the authors prove whether a single deterministic algorithm wins on every instance ("universal solvability").

The authors show: All single-valued cases are solvable (Thm 4); for finite-output maps identification needs non-passive observation of both objects (Thm 1), prediction needs only active candidates (Thm 2), verification sits in between (Thm 3); for infinite-output
maps testing the ground truth becomes necessary (Thm 5). The positive results adapt Kleinberg & Mullainathan (2024) and the impossibility results go back to Gold (1967).

**Strengths**:

The setting is interesting and the authors cover a lot of cases (81 of them), which is nice to have in one place. Their main claim is that what decides feasibility is the type of observation, in particular whether you can get negative information, and this seems like a reasonable way to organize the problem. Some of the results, the impossibility ones especially, are non-trivial.

**Weaknesses:**

1. "Universal solvability" is used in the abstract and throughout the introduction but
is only defined quite late in on page 5. It would be better if at least a simplified definition is added up front (deterministic learner, adversary controls the enumeration, stabilization in the limit, no convergence certificate or sample complexity).

2. The finite-output assumption is essential to every result in the proofs is not present in the theorem statements. May be a clean fix would be to state a single "Assumption 1" (countable X, Y; truth in the candidate class; distinct candidates; the relevant output condition) and write each theorem as "under Assumption 1, ...".

3. I feel authors should be a bit cautious about how strongly the claims are phrased relative to what is assumed (countable spaces, truth in the candidate class, no noise, no time bound). Under such generous assumptions "universal solvability" is a weak guarantee. Essentially, if the answer is in your enumerable list and you eventually see enough, you will find it. The same wording could be used to "prove" almost anything (e.g. that discovering the true laws of physics is universally solvable, since they are single-valued and Theorem 4 applies). I do not think this undermines the results, but the applied examples (AlphaFold, alignment, etc.) are presented without much practical consideration  which makes the claims read as stronger than they really are. Tightening the wording so the idealizations work well with the examples would fix this.

4. The related literature is very short. I feel several directly adjacent areas should be discussed: there might be more, but online / mistake-bound learning (Littlestone, 1988) is closely related, and work from model selection and hypothesis testing could be mentioned as well.

5. Minor: The intro sentence "using data to write down a differential equation to write down a governing equation" reads garbled/duplicated.

**Additional Comments:**

N/A

**Audience:**

Yes

**Audience Explanation:**

Some people in Learning theory might like it.

**Broader Impact Concerns:**

No real ethical concerns; the work is theoretical.

**Claims And Evidence:**

Yes

**Claims Explanation:**

Mostly Yes. But see Weakness 3 above

**Requested Changes:**

See the weaknesses section in the summary.

---

> ### Author Response · Authors · 2026-07-12
>
> We thank the reviewer for the careful review and insightful comments. We appreciate the positive assessment of the question and the systematic organization of the 81 cases. Below we address the weaknesses and requested changes raised in the review.
>
> **W1:** We agree that universal solvability should be understandable before the formal game definitions. The introduction now defines it as the existence of one deterministic learner whose answers are all correct after some finite, instance-dependent time, for every exactly realizable instance and every admissible adversarial enumeration. The revised text also emphasizes that the learner is not told when stabilization occurs and that the result provides no uniform sample, query, or computational bound. The complete formal definitions remain with the corresponding games in Sections 3-5.
>
> **W2:** We have added a single "Standing assumptions" paragraph at the beginning of Section 2 and made the output condition explicit in every theorem. Theorems 1-3 state that the maps are finite-output, Theorem 4 states that the target and candidates are total single-valued maps, and Theorem 5 states the possibly-infinite-output setting. The standing assumptions include countable input and output spaces, a countable candidate list containing the target, noiseless observations, exhaustive fixed enumerations, and exact realizability. We do not require the candidates to be distinct because distinctness is unnecessary: when several candidates equal the target, $j_*$ denotes the least such index, and any index corresponding to the target is a correct identification.
>
> **W3:** We agree that the formal guarantee is deliberately weak compared with a practical learning guarantee, and we now say so prominently. The meaning of universal solvability and its limitations are stated before the results. Section 1.2 explains that the framework provides no finite-sample, computational, optimization, noise-robustness, or model-misspecification guarantee. Appendix A presents every application only as a conditional mapping into the formal game and explicitly identifies assumptions that fail in practice. The abstract and conclusion have also been softened so that the results are described as an information-theoretic guide within the idealized assumptions rather than as unconditional statements about scientific discovery or present AI systems.
>
> **W4:** We have expanded the related-work discussion in Section 1.1 and Appendix B.3. The paper now compares the framework with online mistake-bound learning and explains that, unlike that setting, our learner receives no correctness feedback and is not evaluated through a mistake bound. We also compare with statistical model selection and hypothesis testing, which posit sampling distributions and use likelihood, risk, or finite-sample error criteria. Our games instead use fixed adversarial oracle interactions, exact realizability, and eventual correctness. The revised discussion also covers the adjacent exact-query and language-generation literature.
>
> **W5:** We thank the reviewer for catching the duplicated sentence. It has been replaced by a concise description of map identification as recovering the governing map, such as the governing PDE.
>
> We hope this answers the reviewer's questions and concerns. We are happy to answer any follow-up question(s) that the reviewer may have.

---

### Review · Reviewer_XQcm · 2026-07-06

**Summary Of Contributions:**

The paper asks when an unknown input-output system can be learned in an
information-theoretic, ``in the limit'' sense, that is, whether a learner can
eventually succeed after enough data, without attention to sample size, computation,
or noise. The system is modeled as a set-valued map $M^\star$ that assigns each
input a set of valid outputs. The learner is given a countable list of candidate
maps $M_1,M_2,\ldots$, one of which is guaranteed to be adequate, and must carry
out one of three tasks: identify the candidate equal to $M^\star$, produce a
valid output for a given input, or decide whether a given input has any valid output
and, if so, return one.

The paper considers three observation models, applied separately to the ground
truth and to the candidates: passive observations of input-output pairs, active
queries in which the learner chooses an input and receives a valid output or an
emptiness report, and testing queries in which the learner proposes a pair and is
told whether it is valid. Crossing the three tasks, the three ground-truth
observation models, the three candidate observation models, and three structural
regimes for the maps, single-valued, finite-output, and infinite-output, yields 81
cases, for which the paper gives solvability conditions.

The main strength is that the paper tries to give a systematic taxonomy for a broad
class of input-output learning problems. The main weakness is that, after encoding
input-output pairs as strings, several of the 81 cases of the framework appear very close to classical
language identification in the limit and recent language generation in the limit.
The paper does not sufficiently distinguish which parts of its 81-case taxonomy are
inherited from this prior work and which parts are genuinely new.

**Additional Comments:**

Here is the main content supporting my evaluation.

**Graph-language encoding**

Let $X$ and $Y$ be *countable sets*, and fix an *encoding*

$\langle \cdot,\cdot\rangle:X\times Y\to \Sigma^*$

For every *set-valued map* $M:X\to\mathcal P(Y)$, define the language of its valid
pairs by

$L_M:=\{\langle f,g\rangle:f\in X,\ g\in M(f)\}\subseteq \Sigma^*$

For each input $f$, the corresponding fiber is

$L_{M,f}:=\{g\in Y:\langle f,g\rangle\in L_M\}=M(f)$


This connects directly to the classical *Gold-Angluin* model of language
identification in the limit and to the recent *Kleinberg-Mullainathan* model of
language generation in the limit.

**Definition: Language identification in the limit.** There is an unknown target language $L^\star\subseteq\Sigma^*$ and candidate
languages $L_1,L_2,\ldots$, with $L_j=L^\star$ for at least one $j$. The
learner observes a *positive text* for $L^\star$, and after each finite prefix
outputs an index. It succeeds if its output eventually stabilizes to some $j$
with $L_j=L^\star$.

**Definition: Language generation in the limit.** With the same target and candidate languages, the learner again observes a positive
text for $L^\star$. Instead of identifying $L^\star$, it must eventually output
*valid strings*: from some finite time onward, each produced string lies in
$L^\star$.

In the formulation of Kleinberg and Mullainathan, the generated string is also
required to be *new*, i.e., not already seen in the training text. This novelty
requirement is not needed for the comparison with the paper's map-prediction task,
but only makes the generation problem stronger.

**Definition: Prompted language generation.** For a language $L\subseteq X\times Y$, define the fiber over a prompt $f$ by

$L_f:=\{g\in Y:\langle f,g\rangle\in L\}$

In prompted generation, after observing positive examples from $L^\star$, the
learner is given a *prompt* $f$ with $L^\star_f\neq\emptyset$ and must output
some $g\in L^\star_f$. Success means that, after finitely many rounds, all
prompted outputs are valid.

**Section 1. Passive observations**

A *passive observation* of $M$ is exactly a *positive example* from $L_M$. Thus,
when the ground-truth map is passively observed, the training stream in the paper
is precisely a positive text for the language $L_{M^\star}$.

*Map identification.*

Given candidate maps $M_1,M_2,\ldots$, set $L_i=L_{M_i}$. Since $M_i=M^\star$
if and only if $L_{M_i}=L_{M^\star}$, passive map identification is just language
identification applied to these languages. Conversely, ordinary language
identification is a special case: for a fixed $y_0\in Y$, encode
$S_i\subseteq X$ by $M_i(f)=\{y_0\}$ when $f\in S_i$, and
$M_i(f)=\emptyset$ otherwise.

*Map prediction.*

For a map $M$, the valid outputs for input $f$ are exactly the fiber
$M(f)=L_{M,f}$. Therefore the paper's map prediction task, given $f$ with
$M^\star(f)\neq\emptyset$, output some $g\in M^\star(f)$, is precisely
prompted language generation for $L_{M^\star}$. The unprompted language
generation problem is the special case where the prompt set has one element.

*Map verification.*

The valid-input domain of a map is
$\operatorname{Dom}(M)=\{f\in X:M(f)\neq\emptyset\}$, which is exactly the set of
prompts with $L_{M,f}\neq\emptyset$. Thus map verification is prompted fiber
non-emptiness testing for $L_{M^\star}$, together with witness generation in the
nonempty case.

**Section 2. Active observations**

In the active observation model, the learner chooses an input $f$, and the oracle
returns some $g\in L_{M,f}$ if $L_{M,f}\neq\emptyset$, and reports emptiness
otherwise. Thus active map access is *chosen-prompt positive-completion access*, not
*membership-query access* to $L_M$. It supplies a valid completion, but does not
decide an arbitrary pair $(f,g)$ or certify that a particular $g$ is absent
from $M(f)$.

The correspondences are otherwise unchanged. Active map identification is language
identification of $L_{M^\star}$ with chosen-prompt positive data acquisition.
Active map prediction remains prompted generation over the fibers
$L_{M^\star,f}$, and active map verification remains fiber non-emptiness testing
with witness generation, since the active oracle has exactly this shape on queried
prompts.

**Section 3. Testing observations**

Testing access is exactly *membership-query access* to $L_M$:

$T_M(f,g)=\mathbf 1[g\in M(f)]=\mathbf 1[\langle f,g\rangle\in L_M]$

Thus the paper's testing-testing model, where both the ground-truth and
candidate maps can be queried by tests, gives membership-query access on both
sides: $T^\star$ queries $L_{M^\star}$, while $T_i$ queries $L_{M_i}$.

For identification, this is the *membership-query version* of language
identification. The goal is still to find $i$ with $M_i=M^\star$, equivalently
$L_{M_i}=L_{M^\star}$, and a proposed disagreement can be checked by comparing
the two membership answers on $(f,g)$. Candidate testing alone only gives
information about $L_{M_i}$, while ground-truth testing alone only gives
information about $L_{M^\star}$; equality testing naturally needs access to both
sides.

For prediction, ground-truth testing after $f$ is revealed turns *prompted
generation* into search over $g_1,g_2,\ldots$ until a positive witness is found.
Candidate testing is weaker than active candidate access, since it checks proposed
pairs but does not decide whether a candidate fiber is nonempty or produce a
member. For verification, pairwise testing verifies a witness, but over infinite
$Y$ it does not certify emptiness in finite time. Hence verification remains
*fiber non-emptiness testing* with witness generation.

**Audience:**

Yes

**Audience Explanation:**

At least some of TMLR's audience should be interested in the topic. The paper is
about learnability of input-output systems under different forms of observation, and
this connects naturally to learning theory, language generation, scientific modeling,
and formal accounts of prediction and verification. A clear taxonomy of when
identification, prediction, and verification are possible could be useful.

However, the current version does not yet make the contribution easy to evaluate.
The paper should more explicitly situate its taxonomy relative to language
identification in the limit, language generation in the limit, prompted generation,
membership-query models, feedback/query variants, etc.. With that positioning, the
paper would definitely be interesting to readers who care about foundations of generation and
learnability.

**Claims And Evidence:**

No

**Claims Explanation:**

I do not think the current submission supports its claims with sufficiently clear
evidence. My concern is not mainly with the internal correctness of each stated
theorem, but with the framing and evidence for novelty. Once the encoding is made
explicit, a set-valued map $M$ can be represented by the language of its valid
pairs,

$ L_M=\{<f,g> :g\in M(f)\} $

Under this representation, the problems studied in this work are very similar to the problems studied in the literatures on language identification in the limit and language generation in the limit.
For instance, map identification becomes language identification of
$L_{M^\star}$, prediction becomes prompted generation, and verification becomes
prompt-level non-emptiness testing together with witness generation (more details on this later). These are very
close to previously studied problems in language identification and language
generation in the limit, including the Gold-Angluin line of work and the recent
Kleinberg-Mullainathan model.

I think this is particularly important because the paper's terminology (maps, fibers, etc.) makes the connection to prior work harder to see. As a result, the reader cannot readily tell which cases in the 81-case table are new and
which are restatements or direct variants of known language-learning results. (Several of these appear to have been studied before; more details below.)

The application-level claims may also need some calibration: for instance, the model assumes a
countable list of possible candidate maps. But there may be uncountably many maps
between the input and output spaces, so the paper should specify what information
reduces the space to a countable candidate list and how that information is obtained.
This is especially relevant for the PDE examples: if the maps are described using
real-valued data, then there are typically uncountably many possible maps, so that
application at least needs an explicit caveat.

**Requested Changes:**

Some of the changes that would be good:

C1. **Make the graph-language encoding explicit up front:** State plainly that
every map $M:X\to\mathcal P(Y)$ induces the language

$L_M=\{<f,g>:g\in M(f)\}$

and explain what is new relative to language identification and generation: the
input-output and the separate access models for the ground truth
and the candidates.

C2. **Compare case by case to prior work:** Ideally, add a table that, for
each of the 81 cases, marks whether the result is inherited from Gold/Angluin-style
identification, is a variant of Kleinberg-Mullainathan-style generation, corresponds
to a membership-query or feedback model, or relies on a genuinely new aspect of the
map formulation. This is the single change that would most help the reader judge the
contribution. (E.g., here is a list of recent work on language generation in the limit: https://languagegeneration.github.io/ and here is a book on work on language identification in the limit: https://mitpress.mit.edu/9780262100779/systems-that-learn/)

C3. **Clarify the novelty of the taxonomy:** The paper currently reads as a
large list of solvability verdicts. It should distill the classification into a few
conceptual statements, identifying the structural features that drive solvability,
such as single-valued versus multi-valued maps, finite versus infinite fibers, and
which side has active or testing access.

C4. **Calibrate the countability and application assumptions:** The model
assumes countable $X$, countable $Y$, and a countable candidate list. The paper
should explain why this is the right abstraction, what natural countable
restrictions are intended, and how this interacts with examples such as PDEs, where
continuous domains and real-valued data typically give uncountably many possible
maps.

C5. **Qualify broad application claims:** Claims about universal solvability
of AI text generation, impossibility of AI alignment, or scientific discovery should
be softened unless the modeling assumptions are stated much more explicitly. The
formal results concern idealized countable, realizable, noiseless,
eventually-correct learning games.

C6: **Discuss recent generation variants explicitly:** The paper should
discuss prompted generation, query and feedback models, breadth or exhaustive
generation, hallucination/testing variants, and augmented-observation models. The
prediction task here is one-witness generation and does not address breadth, mode
collapse, representative generation, or distributional correctness.


Changes which are not critical but would strengthen the paper

S1. **State the formal model earlier:** The paper spends considerable space on
applications before the reader knows what the learner observes and what counts as
success. At least an informal version of the Section 2 definitions should precede
the statement of results.

S2. Move some technical content into the main body. All proofs are
currently deferred to the appendix, which makes the contribution hard to assess. At
least the main proof ideas, reductions, and representative algorithms belong in the
main text.

S3. Some minor typos and wording issues:

- Why use $f\in\mathcal X$ and $g\in\mathcal Y$, rather than the more standard
$x\in\mathcal X$ and $y\in\mathcal Y$?

- In Definition 2, the expression
$\ell=\operatorname{mod}(k-1,n)+1$ is not well-defined when $n=\infty$. Since
countably infinite graphs are the natural case, the definition needs separate
finite and infinite cases.

- "Testing" is a somewhat awkward name alongside "passive" and "active":
the latter two are adjectives, while "testing" is a noun/gerund. Consider
"test-based," "oracle," or "membership-query" observation.

- "where one only wants to predict ... but also verify" $\to$ "where one
wants not only to predict ... but also to verify."

- "AI-detection ... is map identification task" $\to$ "is a map
identification task."

- "inserting in more problem-dependent knowledge" $\to$ "inserting more
problem-dependent knowledge."

- "then it universally solvable" $\to$ "then it is universally solvable."

- "one of the candidate models successful achieves" $\to$ "successfully
achieves."

- "spams emails" $\to$ "spam emails."

- "good moves. as in AlphaZero" $\to$ "good moves, as in AlphaZero."

- "$M^\star$ is a indicator" $\to$ "$M^\star$ is an indicator."

- The proof prose in Appendix G is too informal in places, e.g., "I have in
mind" and "I changed my mind."

---

> ### Author Response · Authors · 2026-07-12
> **Official Comment by Authors (1/2)**
>
> We thank the reviewer for the careful review and insightful comments. We especially appreciate the detailed explanation of the connection with language identification and generation. It guided the most substantial changes in this revision. Below we address the requested changes and suggestions raised in the review.
>
> **C1:** We agree that the connection to language learning should be transparent before the reader reaches the results. Section 2 now presents this connection in two concrete ways. First, ordinary language generation is the singleton-input case: for $\mathcal{X}=\{f_0\}$, the outputs $\mathcal{M}_L f_0=L$ are precisely the valid strings in the language. Second, ordinary language identification is the singleton-output case: with $\mathcal{Y}=\{1\}$, we define $\mathcal{M}_L f=\{1\}$ exactly when $f\in L$. Passive observations then give positive language examples, and testing $(f,1)$ is a membership query. Appendix B.1 states the general one-to-one correspondence between a map and the language of its valid input-output pairs. Appendix B.3 further distinguishes active access, which returns a completion or certifies an empty output set for a chosen input, from membership-query access, which tests a proposed pair. The additional structure in our setting is that outputs remain organized by input and that the learner may have different types of access to the target and the candidates.
>
> **C2:** We thank the reviewer for this suggestion. We have added the requested case-by-case comparison in Tables 1 and 2 of Appendix B. Table 1 defines the argument prototypes, states whether each supports a universal algorithm or an impossibility result, gives its relationship to earlier work, and points to the complete proof. In particular, LI, SCD, and MQ identify arguments taken from or adapted from language identification, Kleinberg-Mullainathan generation, and membership-query learning. FA, NI, FI, IO, and CT identify arguments developed for the present input-output setting. SV and Sub[X] identify a direct single-valued argument and lower bounds reused from the finite-output subclass. Table 2 then assigns these codes to all 81 cases. Light red, green, and yellow backgrounds distinguish the three groups, and combined codes show when a conclusion uses more than one ingredient.
>
> **C3:** We have reorganized the presentation around the structural reasons for the classification. The main text now emphasizes four recurring ideas. First, for total single-valued maps, positive observations eventually reveal the unique output, so negative data are unnecessary. Second, for finite-output maps, repeated active observations reveal the entire output set after the outputs begin to repeat and therefore provide negative information. Third, prediction requires only one valid output, whereas identification compares complete maps and verification must also decide whether an output set is empty. Finally, with possibly infinite output sets, observing more outputs never proves that an unseen output is absent, so target membership queries become essential. These ideas are stated in the introduction, developed in the proof discussions after Theorems 1-5, and reflected directly in the argument-prototype tables.
>
> **C4:** We agree that countability of the candidate list is additional prior information and does not follow merely from countability or discretization of the input and output spaces. The introduction now gives concrete examples of countable candidate classes, including finitely represented architectures and finite-precision parameter settings. Section 1.2 collects the limitations and states explicitly that the theory does not range over an unrestricted real-parameter family. We also added an approximation interpretation: at a fixed tolerance $\varepsilon$, outputs can be grouped into bins of diameter at most $\varepsilon$, so exact realization of the discretized map corresponds to $\varepsilon$-accurate prediction at that resolution. Appendix A.7 applies this viewpoint carefully to PDE learning by restricting the forcing terms to a prescribed countable set and the models to finitely described configurations. We also state that the theorem provides neither a sample-complexity bound nor a conclusion about the limit $\varepsilon\to 0$.

---

> ### Author Response · Authors · 2026-07-12
> **Official Comment by Authors (2/2)**
>
> **C5:** We appreciate this caution and have qualified the application claims throughout the paper. The abstract and Section 1.2 now describe the results as idealized, information-theoretic, in-limit statements. Appendix A begins with a global statement that every application is conditional on countable spaces and candidates, exact realizability, noiseless access, exhaustive oracle enumerations, and eventual rather than resource-bounded correctness. Each application then includes its own caveat. For example, the language-model discussion explicitly says that the formal separation between prediction and verification is not a claim that current LLM training universally succeeds or that practical alignment is impossible. Similar qualifications have been added to the protein-design, spam-filtering, theorem-proving, PDE, chess, and speaker-identification examples.
>
> **C6:** Sections 1.1, B.2, and B.3 now compare our task with prompted, uniform, non-uniform, and exhaustive generation; generation with optional feedback; breadth, hallucination, mode collapse, and stability; noise, loss, membership feedback, and augmented observations; and the broader family of exact-query models. We state explicitly that our prediction task asks only for one eventually valid witness. It does not require novelty, exhaustive or representative coverage, calibrated probabilities, distributional correctness, breadth, or noise robustness. We also explain that the prompted result of Kleinberg and Mullainathan uses different prompt and query assumptions, whereas our finite-output prediction algorithm must handle every input with a nonempty target output set, including inputs on which earlier candidates may have no output.
>
> **S1:** The introduction now gives an informal version of the observation model before the formal results. It defines passive, active, and membership-query observations and explains that access to the target and candidates is selected separately. It also defines universal solvability as eventual correctness of one deterministic learner for every admissible, adversarially ordered instance, with no certificate of stabilization and no uniform resource bound. Section 2 then gives the full definitions and collects the standing assumptions.
>
> **S2:** Every theorem in the main text is now followed by its main proof idea. The discussion explains the positive algorithm, why it eventually stabilizes, and the source of the corresponding lower bounds. Theorem 2 receives a more detailed treatment: the paper first explains the smallest-consistent-candidate algorithm and then plays out the ambiguity left by any finite number of negative tests at a fresh input. The detailed applications were moved to Appendix A so that this intuition could be placed directly beside the theorems. The full proofs remain in Appendices C-G.
>
> **S3:** We use $f$ and $g$ to evoke a forcing term and its response or solution, as is common in operator learning. Section 2 now makes this convention concrete through the PDE solution-operator example. Definitions 2 and 3 separately handle finite cyclic enumerations and countably infinite enumerations, so the modulo expression is never applied with $n=\infty$. We also identify testing explicitly as membership-query access, while retaining the shorter word "testing" in figures and tables. We also thank the reviewer for the careful read. We corrected the "not only ... but also" construction, the duplicated governing-equation phrase, "inserting more problem-dependent knowledge," "an inference task," "spam emails," "good moves, as in AlphaZero," "then it is universally solvable," "successfully achieves," and the indicator-map wording. We also replaced the informal first-person phrases in Appendix G with impersonal descriptions of the same compatible-stage construction. These edits do not change the proof logic.
>
> We hope this answers the reviewer's questions and concerns. We are happy to answer any follow-up question(s) that the reviewer may have.

---

### Author Response · Authors · 2026-07-12
**Summary of the Rebuttal Revision**

We appreciate the reviewers' thoughtful comments, which have helped improve this manuscript. In addition to the minor changes detailed in our responses to each reviewer, we have made the following key updates (marked in red in the revised manuscript) during the revision period:

- We have made the relationship with classical language learning explicit. Section 2 gives two concrete ways to view a language as a map-learning problem, while Appendix B explains the general graph-language correspondence and the additional input-output structure considered in our setting.
- We have added a complete case-by-case account of all 81 results. Table 1 defines the main argument prototypes and separates ideas taken from or adapted from earlier work from arguments developed for the present map setting. Table 2 assigns one or more of these argument prototypes to every case, with colors making the three categories easy to distinguish.
- We have stated the formal assumptions and the meaning of universal solvability much earlier. The revised paper now emphasizes countability, exact realizability, noiseless observations, adversarially ordered exhaustive enumerations, eventual correctness, and the absence of a uniform sample, query, or computational bound.
- We have added the main proof idea after every theorem. In particular, the discussion after Theorem 2 now explains both why active candidate observations are sufficient and why finitely many candidate membership queries leave an ambiguity at a fresh input. The original detailed proof constructions remain in Appendices C-G.
- We have substantially expanded the related-work discussion to cover language identification, prompted and exhaustive generation, membership-query and feedback models, hallucination and breadth, online mistake-bound learning, and statistical model selection and hypothesis testing.
- We have moved the detailed applications to Appendix A and qualified them throughout. The revised discussion explains how finitely represented candidate classes and fixed-resolution discretizations can produce countable problems, including an approximation interpretation based on output bins of diameter at most $\varepsilon$.